# Mouse retinal cell behaviour in space and time using light sheet fluorescence microscopy

Claudia Prahst[1†]*, Parham Ashrafzadeh[2†], Thomas Mead[3,4†], Ana Figueiredo[5], Karen Chang[6], Douglas Richardson[7], Lakshmi Venkaraman[1,2], Mark Richards[2], Ana Martins Russo[5], Kyle Harrington[1‡], Marie Ouarné[5], Andreia Pena[5], Dong Feng Chen[6], Lena Claesson-Welsh[2], Kin-Sang Cho[6,8], Claudio A Franco[5], Katie Bentley[1,2,3,4,9]*

[1]Center for Vascular Biology Research and Department of Pathology, Beth Israel Deaconess Medical Center, Harvard Medical School, Boston, United States; [2]The Beijer Laboratory, Department of Immunology, Genetics and Pathology, Uppsala University, Uppsala, Sweden; [3]The Francis Crick Institute, London, United Kingdom; [4]Department of Informatics, Faculty of Natural and Mathematical Sciences, Kings College London, London, United Kingdom; [5]Instituto de Medicina Molecular, Lisbon, Portugal; [6]Schepens Eye Research Institute of Massachusetts Eye and Ear, Department of Ophthalmology, Harvard Medical School, Boston, United States; [7]Harvard Center for Biological Imaging, Department of Molecular and Cellular Biology, Harvard University, Cambridge, United States; [8]Geriatric Research Education and Clinical Center, Office of Research and Development, Edith Nourse Rogers Memorial Veterans Hospital, Bedford, United States; [9]Biomedical Engineering Department, Boston University, Boston, United States

**\*For correspondence:**
Claudia.prahst@gmail.com (CP);
katie.bentley@kcl.ac.uk (KB)

[†]These authors contributed equally to this work

**Present address:** [‡]Virtual Technology and Design, University of Idaho, Moscow, United States

**Competing interests:** The authors declare that no competing interests exist.

**Abstract** As the general population ages, more people are affected by eye diseases, such as retinopathies. It is therefore critical to improve imaging of eye disease mouse models. Here, we demonstrate that 1) rapid, quantitative 3D and 4D (time lapse) imaging of cellular and subcellular processes in the mouse eye is feasible, with and without tissue clearing, using light-sheet fluorescent microscopy (LSFM); 2) flat-mounting retinas for confocal microscopy significantly distorts tissue morphology, confirmed by quantitative correlative LSFM-Confocal imaging of vessels; 3) LSFM readily reveals new features of even well-studied eye disease mouse models, such as the oxygen-induced retinopathy (OIR) model, including a previously unappreciated 'knotted' morphology to pathological vascular tufts, abnormal cell motility and altered filopodia dynamics when live-imaged. We conclude that quantitative 3D/4D LSFM imaging and analysis has the potential to advance our understanding of the eye, in particular pathological, neurovascular, degenerative processes.

## Introduction

Eye diseases, such as diabetic retinopathy, age-related macular degeneration, cataracts, and glaucoma are becoming increasingly common with the increased age of the general population. Although advances in understanding and treating eye diseases have been made, the cellular and molecular mechanisms involved are still not fully understood. We believe that is partially due to the inadequate ability to image eye tissue in its natural, spherical state, to reveal the many distinct layers with interacting cell types oriented differentially within or between the layers. Optical coherence

**eLife digest** Eye diseases affect millions of people worldwide and can have devastating effects on people's lives. To find new treatments, scientists need to understand more about how these diseases arise and how they progress. This is challenging and progress has been held back by limitations in current techniques for looking at the eye. Currently, the most commonly used method is called confocal imaging, which is slow and distorts the tissue. Distortion happens because confocal imaging requires that thin slices of eye tissue from mice used in experiments are flattened on slides; this makes it hard to accurately visualize three-dimensional structures in the eye.

New methods are emerging that may help. One promising method is called light-sheet fluorescent microscopy (or LSFM for short). This method captures three-dimensional images of the blood vessels and cells in the eye. It is much faster than confocal imaging and allows scientists to image tissues without slicing or flattening them. This could lead to more accurate three-dimensional images of eye disease.

Now, Prahst et al. show that LSFM can quickly produce highly detailed, three-dimensional images of mouse retinas, from the smallest parts of cells to the entire eye. The technique also identified new features in a well-studied model of retina damage caused by excessive oxygen exposure in young mice. Previous studies of this model suggested the disease caused blood vessels in the eye to balloon, hinting that drugs that shrink blood vessels would help. But using LSFM, Prahst et al. revealed that these blood vessels actually take on a twisted and knotted shape. This suggests that treatments that untangle the vessels rather than shrink them are needed.

The experiments show that LSFM is a valuable tool for studying eye diseases, that may help scientists learn more about how these diseases arise and develop. These new insights may one day lead to better tests and treatments for eye diseases.

tomography (OCT) is an established medical imaging diagnostic tool that uses light to capture micrometre-resolution, three-dimensional images, non-invasively (*Srinivasan et al., 2006*; *Huber et al., 2009*). Its main strength lies in revealing information on tissue depth preserving the eyes natural state. However, its limitation lies in not being able to provide a wide field of view, cellular or molecular information. Furthermore, being a non-fluorescent method, specific proteins cannot be labelled and tracked to investigate mechanisms. Currently, only confocal microscopy can deliver this detailed fluorescently labelled information (*del Toro et al., 2010*), but the 3D nature of the tissue is likely distorted during flat-mounting and it is currently not known to what extent this might impact the obtained results. For instance, the vascular biology field is one clear example where these limitations can have a substantial impact. The mouse retina is a common model used to study vascular development and disease; confocal imaging approaches have been used to measure vessel morphology, vascular malformations, junctional organisation, and pathological tuft formation (*Gerhardt et al., 2003*; *Bentley et al., 2014*; *Stahl et al., 2010*). Moreover, vessel diameters are now being used to predict blood flow (*Bernabeu et al., 2014*; *Baeyens et al., 2016*). Distortions arising from confocal flat-mounting could therefore have important ramifications for the overall conclusions of several studies.

Changes in cellular and tissue morphology are a hallmark of many eye diseases. For instance, retinopathy of prematurity and diabetic retinopathy are characterised by excessive, bulbous and leaky blood vessels that protrude out of their usual layered locations. These malformed vessels cause many problems including the generation of abnormal mechanical traction, which pulls on the different layers, eventually leading to detachment of the retina (*Nentwich and Ulbig, 2015*; *Hartnett, 2015*). Yet, very limited information has arisen on the conformation and morphogenesis mechanisms of these vascular tuft malformations, despite a wealth of confocal studies of the related oxygen- induced retinopathy (OIR) mouse model (*Connor et al., 2009*).

Another limitation of confocal microscopy for imaging of mouse retinal angiogenesis is the inability to perform live imaging of endothelial cell dynamics. Endothelial cells move and connect in highly dynamic, complex ways to generate the extensive vascular networks required to perfuse the retina over time (angiogenesis). Live, in vivo imaging of murine intraocular vasculature has been reported using confocal microscopy (*Ritter et al., 2005*) and holds great promise for dynamic longitudinal

studies of the growth/regression of large vessel such as hyaloid vessels. However, it does not as yet suit studies of smaller more dynamic cell and subcellular structures as being reliant on confocal currently limits such studies to slow frame rates (5–10 min intervals), limited z stack resolution with photobleaching issues and an apparent limited field of view. There are a small number of reports on ex vivo live-imaging of the retinal vasculature with confocal microscopy, but which clearly entails challenges as dissection of the retina for culture is time consuming, and moreover, the flatmounting is likely to disturb local tissue arrangement and mechanics (*Sawamiphak et al., 2010*; *Rezzola et al., 2013*). Furthermore, photobleaching, phototoxicity and long acquisition times continue to remain an issue.

A growing number of reports show that neurovascular interactions in the eye are important during development and disease progression (*Akula et al., 2007*; *Narayanan et al., 2014*; *Nentwich and Ulbig, 2015*; *Usui et al., 2015*; *Verheyen et al., 2012*). Neurons and vessels are however currently imaged with physical sectioning of paraffin or cryo-embedded retinas, which precludes concurrent visualisation of the vasculature, due to the orthogonal arrangement of neurons and vessels within or between retinal layers respectively. Likewise, current methods have limited potential for quantitative 3D and live imaging of fluorescently labelled neurons in neurodegenerative mouse models.

Recent advances in light-sheet fluorescence microscopy (LSFM) have demonstrated its strength for allowing the rapid acquisition of optical sections through thick tissue samples such as mouse brains (*Stelzer, 2015*). Instead of illuminating or scanning the whole sample through the imaging objective, as in wide-field or confocal microscopy, the sample is illuminated from the side with a thin sheet of light. Thus, in principle LSFM would require little interference with the original spherical eye tissue structure, avoiding distortion of the tissue with flat-mounting. Moreover, LSFM is becoming a gold-standard technique to perform live-imaging in whole organs/organisms because it permits imaging of thick tissue sections without disturbing the local environment, while also reducing photobleaching and phototoxicity (*Stelzer, 2015*; *Reynaud et al., 2015*). Thus, here we investigate the feasibility, advantages and disadvantages of LSFM for imaging the mouse eye for development or disease studies. We present an optimised LSFM protocol to rapidly image neurovascular structures, across scales from the entire eye to subcellular components in mouse retinas. We investigate the pros and cons of LSFM imaging of vessels over standard confocal imaging techniques in early mouse pup retinas. Importantly, we also demonstrate the benefits of LSFM using the OIR mouse model, where we discover previously unappreciated new spatial arrangements of endothelial cells in the onset of vascular tuft malformations due to the improved undistorted, 3D and 4D imaging capabilities of LSFM.

We conclude that LSFM quantitative 3D/4D imaging and analysis has the potential to advance our understanding of healthy and pathological processes in the eye, with a particular relevance for the vascular and neurovascular biology fields, as well as ophthalmology.

## Results

### LSFM enables rapid 3D imaging of mouse eyes, and in particular retinas, in their natural state

To visualise the retinal vasculature using epifluorescence or confocal microscopes, the retina is flatmounted by making four incisions before adding a cover slip containing mounting medium onto glass slides (*Figure 1a*, upper panel). To image samples using LSFM, however, samples are suspended in their natural state in low-melting agarose (*Figure 1a*, lower panel). This enables imaging of the vasculature of large and intact samples such as the whole eyeball (minus the sclera and cornea) (*Figure 1b*), the iris (*Figure 1c*), or the optic nerve (*Figure 1d*). Using LSFM, it was possible to observe the superficial, intermediate and deep vascular plexus (*Figure 1e* and *Figure 1—figure supplement 1b*) of a retina in its native conformation (*Figure 1f*). Acquiring a stack of the entire retina using LSFM contains 200–300 images, still, yet the imaging time is much shorter than it would be using confocal (~1 min).

Imaging the iris microvasculature (*Figure 1c*) revealed that the vasculature network is immature at P15 (*Figure 1b*, *Figure 1—figure supplement 1a*), and that it remodels into a mature network in adulthood (*Figure 1c*). A network of capillaries was visible at P15, whereas the adult

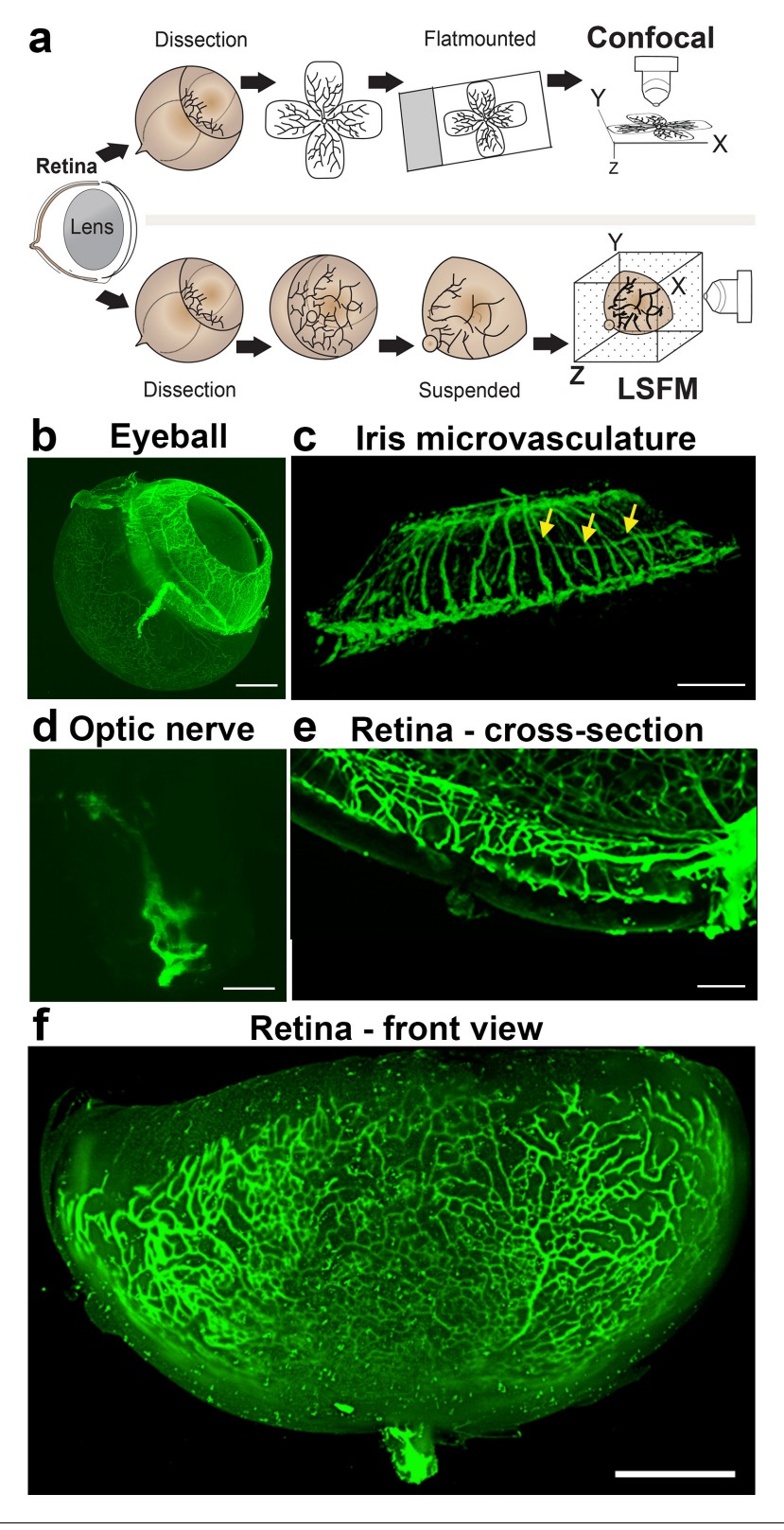

**Figure 1.** Imaging of the whole eye using light sheet microscopy. (a) Schematic of retina preparation for imaging. For conventional confocal microscopy, four incisions are made to enable flat-mounting of the retina onto a cover slip. For LSFM, pieces of the retina are suspended and imaged from a right angle. (b) Maximum intensity projection (MIP) of a P15 mouse eyeball (z = 274 slices). Vessels were visualised with IsoB4 staining. Scale bar, 500

*Figure 1 continued on next page*

*Figure 1 continued*

µm. (c) 3D-rendered image of the adult iris microvasculature (z = 263 slices). Vessels were visualised with IsoB4 (yellow arrows). Scale bar, 250 µm. (d) MIP of the optic nerve. Vessels were visualised with IsoB4 staining (z = 176 slices). Scale bar, 50 µm. (e) MIP of a cross section of a P10 mouse retina. Vessels were visualised with IsoB4 staining. Scale bar, 100 µm. (f) MIP of a whole P10 mouse retina suspended and imaged intact (z = 176 slices). Vessels were visualised with IsoB4 staining. Scale bar, 500 µm. See *Figure 1—figure supplement 1* for additional images.

The online version of this article includes the following figure supplement(s) for figure 1:

**Figure supplement 1.** Iris microvasculature and Golgi-polarity as revealed by LSFM.

---

microvasculature consisted of radial branches of small vessels and capillaries in a relatively linear pattern. The major arterial circles (MICs) around the iris root were developed in both P15 and adult mice (*Figure 1c* and *Figure 1—figure supplement 1a*). The images generated from adult mice using LSFM are consistent with a previous report using OCT to image the iris microvasculature (*Choi et al., 2014*). Using LSFM, the vessels appeared straighter, and the MICs were not as close to the iris root, which could be because OCT involves live-imaging of the vasculature, without mechanically removing the sclera and cornea.

We next tested whether LSFM could resolve subcellular structures in the retinal vasculature such as the Golgi apparatus, which has recently been shown to be important for inferring cell polarity during vessel regression (*Franco et al., 2015*). We found it feasible to stain and image the Golgi organelle (Golph4, Alexa 647) and the collagen IV-containing basement membrane around the vessels (*Figure 1—figure supplement 1c,d*). Moreover, quantification of the nucleus-Golgi polarity axis was amenable when imaging the GNrep mouse (*Barbacena et al., 2019*), which expresses Golgi-localised mCherry and nucleus-localised eGFP upon Cre-mediated recombination, enabling visualisation of endothelial specific nuclei and Golgi apparatus. We measured the polarity of cells in 3D by drawing lines from the centre of their nuclei to the nearest Golgi body. We observed endothelial cells collectively polarising against the flow direction along an arterial network (*Figure 1—figure supplement 1e*), as previously described (*Franco et al., 2016*). The ability to 3D rotate the undistorted vascular image stacks obtained with LSFM revealed hidden cells whose polarity could be analysed, not visible when analysing the same image stack using standard confocal 2D imaging (i.e. viewed only from above) (*Figure 1—figure supplement 1f–h*). On static images of mice expressing Lifeact-enhanced green fluorescent protein (EGFP) (*Riedl et al., 2010*) we performed deconvolution to reduce the light scattering effects and found this gave a marked improvement to the resolution of actin bundles within endothelial cells (*Figure 1—figure supplement 1i–l*). Taken together, we concluded that LSFM can rapidly generate 3D images of the murine eye in its native form across scales, with tissue, cellular and subcellular resolution.

## LSFM enables concurrent 3D imaging of retinal cell types within and between the retinal layers

Neurons are currently imaged by making vertical sections, orthogonal to the three vascular layers (the superficial, intermediate and deep plexus) (*Figure 2a,b*, 'side view'), which necessarily means losing the ability to observe vascular branching in the horizontal layer in the same tissue. Likewise, studies focused on the retinal vasculature use whole mount images of the retina viewed from above (*Figure 2b*, 'top view') using horizontal optical sections (*Usui et al., 2015*), which does not allow proper imaging of retinal neurons spanning between the layers because of insufficient z-resolution in confocal microscopy. Thus, we next investigated whether concurrent imaging of neurons and vessels in the same sample might be achieved with the optical sectioning and rotational viewing capacity of LSFM.

We found that eye cups from P3 C57BL/6 Thy1-YFP mice, labelling retinal ganglion cells in yellow combined with IsolectinB4 labelled vasculature provided 3D high resolution images without the need for tissue clearing (*Figure 2c,d*, *Video 1*). However, we found that including lipid removal/permeabilisation as part of a full tissue clearing protocol further improves resolution for eye cups at later stages of development, when more of the retinal vascular layers have formed (*Figure 2e–h*), as it decreases the scattered light caused by imaging thicker tissue with the light sheet (*Richardson and*

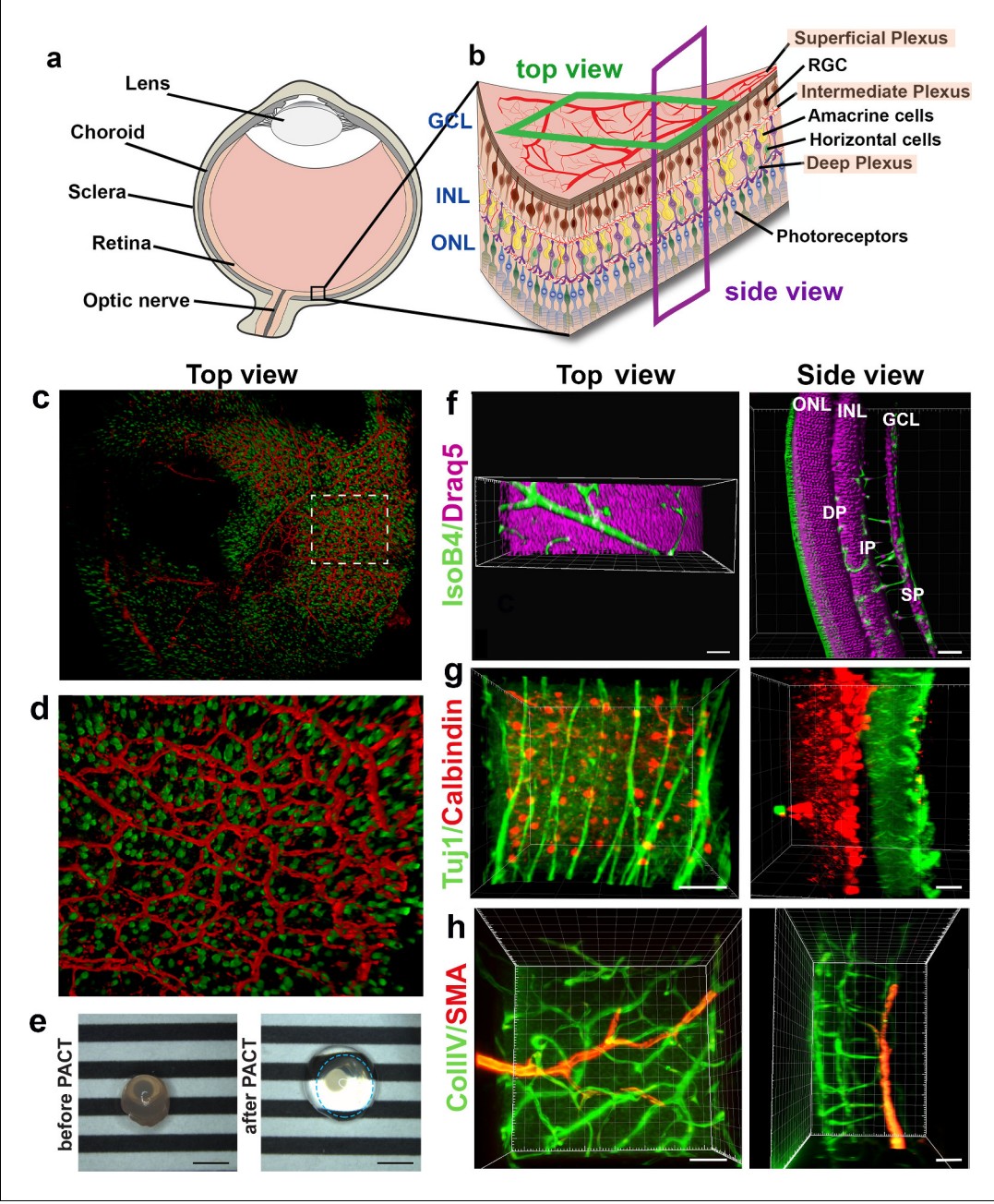

**Figure 2.** 3D reconstruction of nerves and vessels in one image. (**a**) Schematic of an eyeball. (**b**) Schematic of the retina and its cell types. (**c**) Retinal eye cups expressing yellow fluorescent protein, YFP (green) were harvest from Thy1-YFP mice and stained with Isolectin IB4 (red). The retinal eye cups were mounted and imaged with LSFM. (**d**) Enlarged region of c. (**e**) Representative image of an eyeball before clearing (left panel), and an eyeball after PACT clearing (right panel). The circle around the cleared eyeball depicts the outline of the eyeball. Scale bar, 2 mm. (**f**) Draq5 staining (magenta) visualises the inner nuclear layer (INL) and outer nuclear layer (ONL) of the adult mouse PACT cleared retina. Vessels were visualised by IsoB4 staining (green). (**g**) Tuj1 (green) and calbindin (red) visualise the ganglion and horizontal cells in the mouse PACT cleared retina. (**h**) Smooth muscle actin (SMA, red) and Collagen IV staining (Coll.IV, green) visualise the three vascular layers and smooth muscle cells in the mouse PACT cleared retina. Scale bars, 50 µm.

The online version of this article includes the following source data and figure supplement(s) for figure 2:

**Figure supplement 1.** Changes in neuronal density in Rho KO degeneration model.

**Figure supplement 1—source data 1.** Excel file containing source data pertaining to *Figure 2—figure supplement 1c and d*.

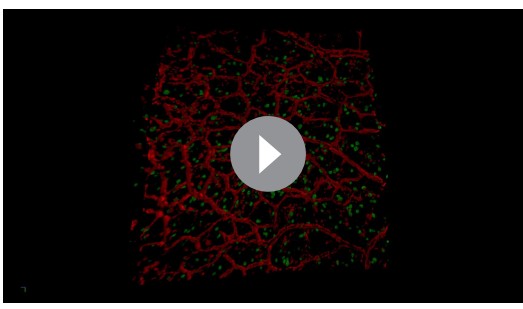

**Video 1.** 3D-rendered LSFM z-stack of retinal eye cups expressing yellow fluorescent protein, YFP (green) harvest from Thy1-YFP mice and stained with Isolectin IB4 (red).
https://elifesciences.org/articles/49779#video1

*Lichtman, 2015*). In order to establish whether LSFM could be used to quantify neuronal changes in a retinal degeneration model we imaged retinal cups from the Rho KO degeneration model (*Figure 2—figure supplement 1*; *Humphries et al., 1997*). The LSFM images were easily segmented and quantification showed a significant decrease in neuronal density in the outer nuclear layer (ONL) at 4 weeks for Rho KO compared to control retinas, which worsened in the 8 weeks Rho KO (*Figure 2—figure supplement 1c*). Measuring ONL thickness showed no notable difference between control and Rho KO at 4 weeks, however, there was a significant reduction in thickness in the Rho KO at 8 weeks relative to both strains at 4 weeks (*Figure 2—figure supplement 1d*). The ONL had almost entirely lost its stable convex curvature by 8 weeks in the KO retina and the inner nuclear layer (INL) also appeared ruffled when viewed in 3D which may be due to the unevenness of dropout of photoreceptors (*Figure 2—figure supplement 1a,b*).

We tested several different clearing methods to see which was better suited to retinal tissue. Using the aqueous-based clearing methods ScaleA2 and FRUIT (*Hou et al., 2015*; *Hama et al., 2011*) did not result in higher quality images and made tissue-handling very difficult during imaging due to the high viscosity of the FRUIT clearing agent. We also tested the passive aqueous-based methods CUBIC-R (*Kubota et al., 2017*) and PROTOS (*Murray et al., 2015*), but again found little improvement. Since many studies use animals genetically engineered to express fluorescent markers such as Tomato or GFP, we decided not to pursue solvent-based clearing methods such as iDISCO, which do not maintain fluorescent protein emission for more than a few days after the clearing process (*Renier et al., 2014*). Overall, we found PACT was the most efficient and effective clearing method for retinal tissue, likely because it is relatively thin (*Yang et al., 2014*; *Treweek et al., 2015*). PACT cleared adult retinas with Draq5 staining, which stains all nuclei, visualising the INL and ONL (*Figure 2f*, *Video 2*). The deep vascular plexus, visualised by IsolectinB4 staining could be seen between the ONL and INL, whereas the intermediate vascular plexus bordered the INL as expected. The superficial vascular plexus is located on the inner retinal surface together with nuclei of the retinal ganglion cells (*Figure 2f*, *Video 2*). Adult retinas were co-immunostained for Tuj1 and Calbindin, markers for retinal ganglion cells and horizontal cells, respectively. This immunostaining made it

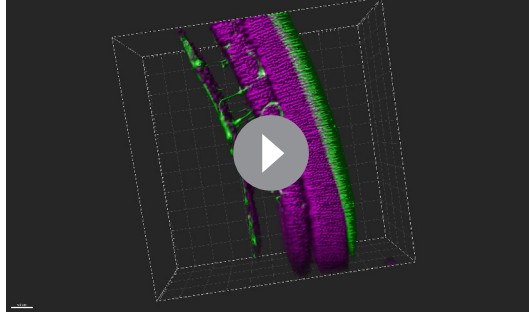

**Video 2.** 3D-rendered LSFM z-stack of all nuclei (Draq5, magenta) visualises the inner nuclear layer and outer nuclear layer of the adult mouse retina. Vessels were visualised by IsolectinB4 staining (green). The retinal pigment epithelium emits green autofluorescence.
https://elifesciences.org/articles/49779#video2

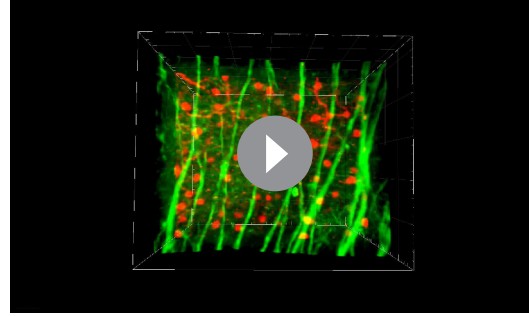

**Video 3.** 3D-rendered LSFM z-stack of an adult mouse retina stained for Tuj1 (green) and horizontal cells (calbindin, red) visualising the ganglion cells and horizontal cells, respectively.
https://elifesciences.org/articles/49779#video3

possible to appreciate the distance between these two cell types in the fully developed retina (*Figure 2g*, *Video 3*). 3D-rendered images of co-staining for smooth muscle actin and collagenIV moreover showed arteries of the superficial vascular plexus covered with smooth muscle cells (*Figure 2h*, *Video 4*). Overall, LSFM holds great promise for concurrent studies of how different cell types interact during eye development and disease.

## Vessel distortion due to confocal flatmounting revealed by correlative LSFM-confocal imaging

As vascular measurements taken from confocal images are used as the standard for inferring the actual sizes of vascular structures in the retina, we next aimed to systematically quantify the 3D distortion of vascular structures incurred by flat-mounting and confocal imaging. In order to make direct, quantitative comparisons of the relatively small vessels in the superficial plexus, we used a correlative LSFM-confocal approach: we first imaged the retinal tissue with LSFM, which retains the natural tissue curvature, then we melted the agarose gel and flat-mounted the same retina onto a coverslip and imaged it again using confocal microscopy (*Figure 3a*). We first analysed the largest vessels near the optic nerve and then smaller capillaries in the sprouting vascular front from P4 WT retinas. Images obtained with our correlative LSFM-confocal approach were then brightness/contrast adjusted and cropped and surface rendered using Imaris to focus on small regions of same vessel segments in the corresponding confocal and LSFM images. Dramatically shallower side views and cross-sections of vessels were evident in the confocal images compared to LSFM (*Figure 3b*). We next quantified this shift in aspect ratio by measuring the diameter taken across the vessel in XY (hereafter 'width') and down through the Z-axis (hereafter 'depth') in the confocal (*Figure 3c*). For LSFM images, given the tissue can be at any orientation in the agarose with respect to the objective, the XYZ coordinate system of the image stack is not indicative of the equivalent width/depth measurement in confocal. Instead, the orientation of the surrounding vascular plexus at the point of the vessel segment was used as a reference surface 'plexus plane' to make the corresponding 'width' diameter measurement, as it is equivalent to the XY plane in the corresponding confocal image. Similarly, the 'depth' diameter in LSFM was defined as perpendicular to the plexus plane and width measurement (equivalent to the diameter through the z-stack in confocal). Vessels were significantly more elliptical (wider and shallower) under the confocal than LSFM, indicative of being compressed during flat-mounting (*Figure 3d,e*). Overall, vessels from retinas flat-mounted for confocal displayed significant distortion, and not in a simple ratio of depth to width changes, indicating LSFM as more reliable for quantitative 3D morphometric studies.

## LSFM enables 4D live-imaging with subcellular resolution, revealing rapid, transient 'kiss and run' tip cell adhesions at the sprouting front

Ex vivo live-imaging could be a useful tool to study tip cell guidance during the angiogenic sprouting process in the mouse retina, but it has proven to be challenging with conventional microscopy. Existing ex vivo confocal methods to live-image retinal vasculature, tissue handling leads to damage of the tissue, as it involves either flat-mounting the retinas onto a membrane and then submerging it in medium (*Sawamiphak et al., 2010*), or cutting the retina into fragments and embedding them in fibrin gels prior to imaging (*Rezzola et al., 2013*).

We therefore established a protocol for live-imaging of the growing retinal vasculature in ex vivo prepared retinas using LSFM. We first crossed mT/mG mice with Cdh5(PAC)-CreERT2 mice and injected them with tamoxifen to induce endothelial GFP expression (*Muzumdar et al., 2007*; *Wang et al., 2010*). Surprisingly,

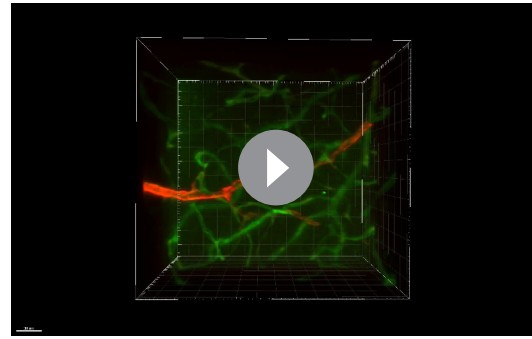

**Video 4.** 3D-rendered LSFM z-stack of an adult mouse retina stained for Smooth Muscle Actin (SMA, red) and CollagenIV staining (CollIV, green) visualises the three vascular layers and smooth muscle cells in the mouse retina.

https://elifesciences.org/articles/49779#video4

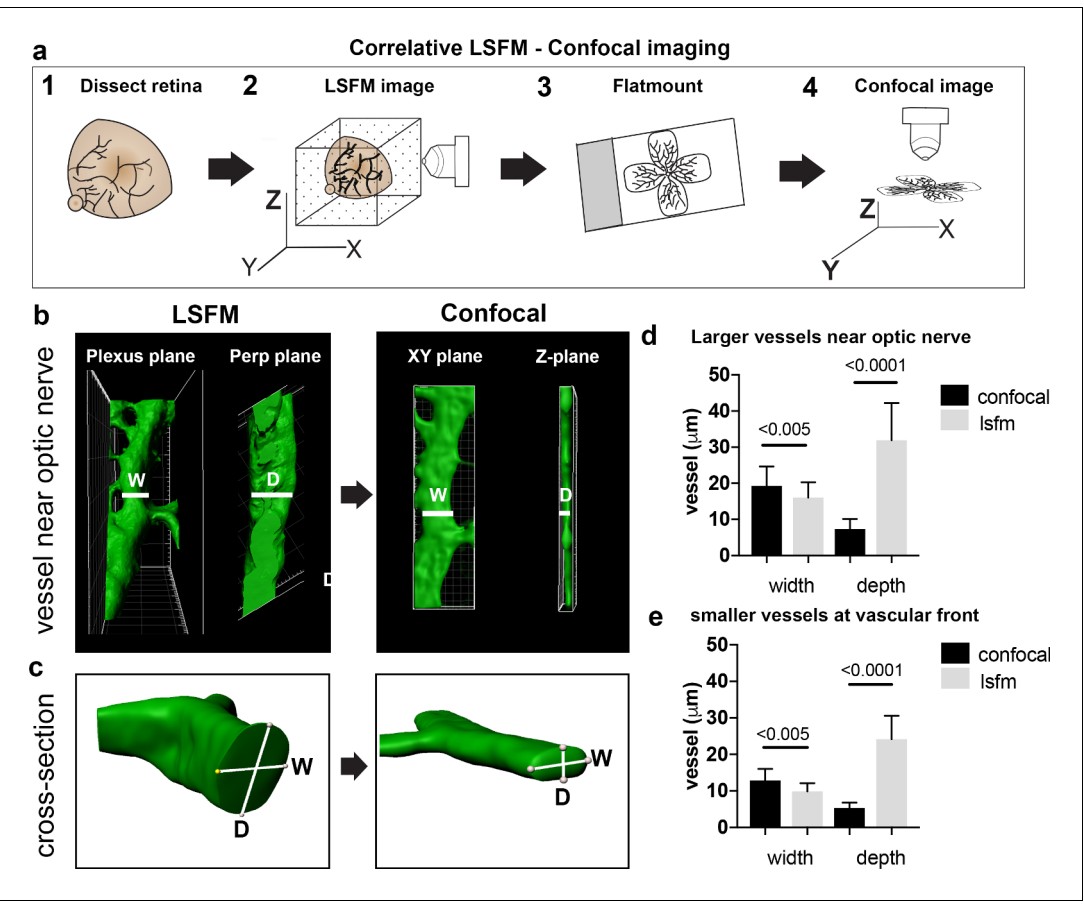

**Figure 3.** Vessel depth distortion in confocal due to flatmounting. (**a**) Schematic showing the correlative LSFM-Confocal imaging approach used to quantify vessel distortion incurred by flatmounting. (**b**) The same large vessel segment imaged first with LSFM then confocal (surface rendered in Imaris). By orienting with the surrounding vessel connections to determine the plexus plane (equivalent to the XY plane in confocal) and the plane perpendicular to it ('perp plane'), which is equivalent to the Z plane in confocal, comparative width (W) and depth (D) measurements can be made of the same vessel segment. (**c**) Cross sectional views of another representative large vessel near the optic nerve shows how the aspect ratio of W and D is shifted to an ellipse in confocal. Near Optic: n = 60 vessels from six retinas (seven images). Vascular Front n = 28 vessels in from four retinas (four images) for each confocal and LSFM.

The online version of this article includes the following source data for figure 3:

**Source data 1.** Excel file containing source data pertaining to *Figure 3d and e*.

connections between ECs formed very rapidly (within 20 min) and regressed just as rapidly (*Figure 4a*, *Video 5*). Such transient 'kiss and run' adhesion and release style interactions between ECs (as opposed to full adhesions or anastomoses, where the connections stably remain) have only been previously reported in glycolysis-deficient ECs in vitro (*Schoors et al., 2014*). The dynamics in vivo were assumed to be slower and more stable than in vitro live-imaging, however our new ex vivo observations indicate a very different set of dynamics and inter-cellular behaviors may be at work in the complex in vivo tissue. Timing is crucial, as the VEGF gradient dissipates after the retinas are dissected and submerged in agarose, at room air and the tissue is therefore no longer hypoxic. However, the directed growth of the filopodia towards the vascular front in our LSFM Videos suggests that this gradient remains intact for at least the first few hours after dissection. Further back from the sprouting front, in the vascular plexus (*Figure 4b*, *Video 6*), we occasionally observed the formation of connections over the course of a few hours, however, branch formation was a rare occurrence. Notably, we did not observe EC apoptosis under these imaging conditions indicating conditions are viable.

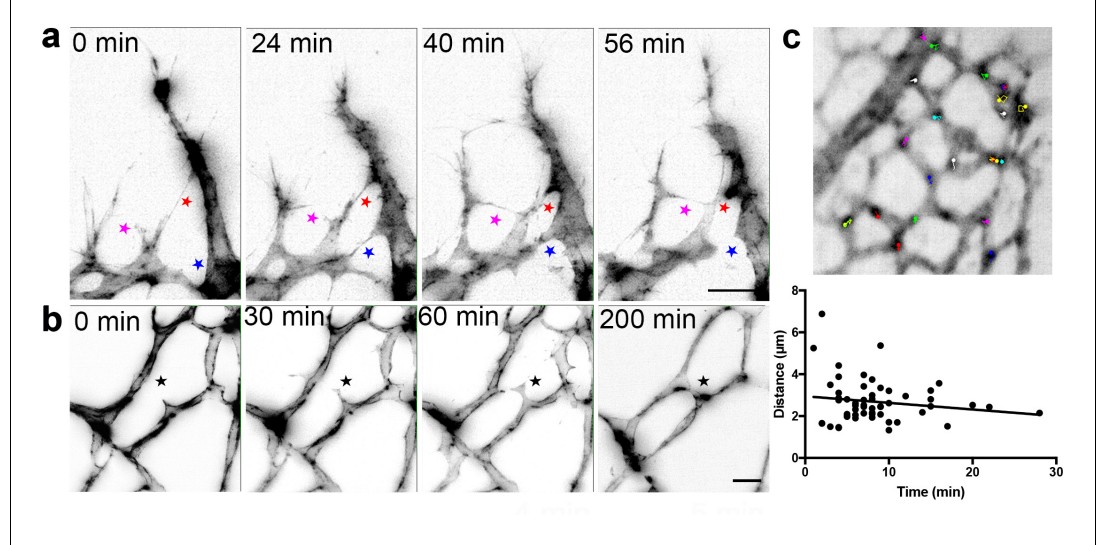

**Figure 4.** Live-imaging of the retinal vasculature. (**a**) Single maximum intensity projections (MIP) of an hour time lapse Video show long, slender filopodia, and rapid fusion and disconnection of tip cells at the vascular front of mT/mG x *Cdh5* (PAC) CreERT2 mice (stars). (**b**) MIPs of a time lapse Video reveal the connection between two branches in the capillary plexus (star). (**c**) lifeAct-EGFP mouse retina at P4/5 were live imaged for 40 min with an interval of one minute per frame. Actin-rich bundles were tracked manually using ImageJ/Fiji. Each color represents one bundle trajectory tracked over time, scale bar is 10 µm. Plot (below) shows each actin bundle's distance travelled over time, average speed was 2.56 µm/min, n = 6 retinas (all uncleared).

The online version of this article includes the following source data for figure 4:

**Source data 1.** Excel file containing source data pertaining to *Figure 4c*.

We next assessed the feasibility of using LSFM to live-image intracellular processes in ex vivo prepared retinas. We dynamically imaged Lifeact-EGFP mice (*Riedl et al., 2010*) with LSFM and quantified the movements of actin-enriched bundles within the endothelial cell bodies in the sprouting front during developmental angiogenesis. Quantitative subcellular actin live-imaging was found feasible (n = 6 retinas) with the average distance travelled by each bundle found to be 2.56 µm (*Figure 4c*, *Videos 7*, *8* and *9*).

Taken together, our LSFM permits the visualisation in real-time of cellular movements with subcellular resolution in the mouse retina, with minimal distortion.

## Three subclasses of pathological retinal neovascular tufts revealed with LSFM

We next sought to image vessels that have grown pathologically, in order to determine whether this imaging method could be used to gain greater insights into eye disease. To this end, we used the OIR model, where mouse pups are placed in 75% oxygen from P7 to P12, and are then kept at room air from P12 to P17 (*Connor et al., 2009*). During the hyperoxia phase, the vasculature regresses, and in the subsequent normoxia phase, new vessels grow in an abnormally enlarged and tortuous manner (*Connor et al., 2009*). Furthermore, vessels also start to grow into the vitreal space forming bulbous vessels, known as 'vascular tufts', above the superficial vascular layer (*Figure 5a*). In the past, it has been difficult to analyse and characterise the growth of these tufts because they are large formations, which appear to be distorted by the flat-mounting process. By performing IsolectinB4 and ERG immunostaining to visualise endothelial cells (ECs) and their nuclei, we obtained 3D-reconstructions of the tufts and were able to first classify them into different groups by quantifying both volume and number of nuclei (*Figure 5a,b*). As expected, we found that the number of nuclei increased with the size of the tuft ($R^2$ = 0.83). Interestingly, however we found many small tufts, and only very few large tufts. The smallest tuft we could identify had two nuclei parallel to each other, the cells growing straight up into the vitreous (*Figure 5a*, upper panel, *Video 10*). We found that most tufts have between 4 and 20 nuclei ('Medium tufts', *Figure 5a*, second panel row, *Video 11*).

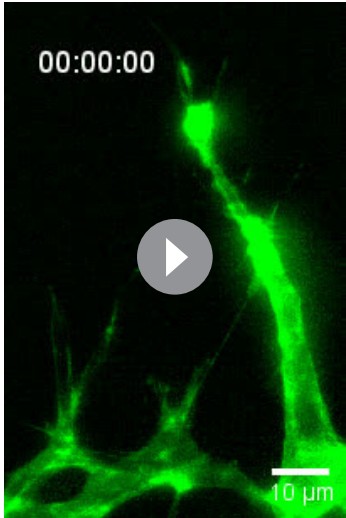

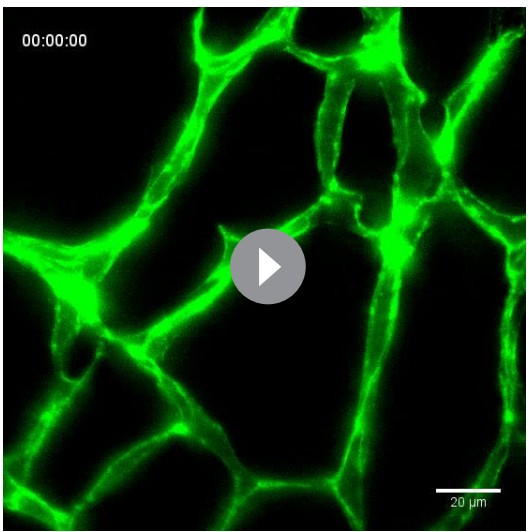

**Video 5.** An hour time lapse LSFM Video showing long, slender filopodia, and rapid 'kiss and run' fusion and disconnection of tip cells at the vascular front of mT/mG x cdh5 (PAC) CreERT2 mice. Frame rate: one image/45 s.

https://elifesciences.org/articles/49779#video5

**Video 6.** A 9 hr time lapse LSFM Video showing a connection between two branches in the capillary plexus of mT/mG x cdh5 (PAC) CreERT2 mice. Frame rate: one image/20 min.

https://elifesciences.org/articles/49779#video6

We also identified a few very 'large tufts' with over 20 nuclei (*Figure 5a*, third panel row, *Video 12*). Next, we quantified the number of connections between the vasculature and the tuft (*Figure 5b*). The large tufts had a higher number of connections to the existing vasculature ($R^2$ = 0.61), Within the medium tuft class there is a linear increase of volume with nuclei number up to approximately 10 nuclei per tuft, but then the volume remained constant despite a doubling of the nuclei number to 20 at the top of this class. Within the large tuft class the volume remained unchanged despite a three-fold increase in nuclei (*Figure 5b*). Intriguingly, the number of connections to the plexus was approximately constant despite the increasing number of nuclei within these classes (*Figure 5c*). However, the number of connections and tuft volume transitioned sharply, to ~2.5 fold and ~3 fold respectively, when the number of nuclei in the tuft exceeded twenty. This indicates that proliferation or an influx of cells to the tuft does not increase tuft volume, but rather, tuft volume only significantly increases when the number of connections to the plexus also increases. Based on this observation, we propose that large tufts are in fact formed by fusion of 2 or three medium tufts.

We observed that some of the vascular tufts contained highly curved nuclei (*Figure 5a*, fourth panel row, yellow arrow, *Video 13*). Quantification of the number of curved nuclei/total nuclei in a tuft showed that in small and medium tufts, the number of curved nuclei correlated well with the number of total nuclei (*Figure 5d*). In large tufts (over 20 total nuclei), the number of curved nuclei

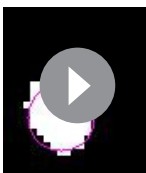

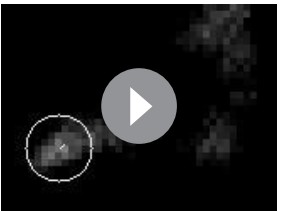

**Video 7.** Representative tracking of a short-lived actin-rich bundle in life-Act-EGFP retina mice imaged with LSFM (7 min). The tracking was performed manually using Manual Tracking plugin in Fiji/ImageJ.

https://elifesciences.org/articles/49779#video7

**Video 8.** Representative tracking of a longer-lived actin-rich bundle in life-Act-EGFP retina mice imaged with LSFM (30 min). The tracking was performed manually using Manual Tracking plugin in Fiji/ImageJ.

https://elifesciences.org/articles/49779#video8

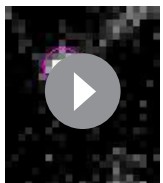

**Video 9.** Representative tracking of a long-lived actin-rich bundle track in life-Act-EGFP retina mice imaged with LSFM (40 min). The tracking was performed manually using Manual Tracking plugin in Fiji/ImageJ.
https://elifesciences.org/articles/49779#video9

was stable suggesting actually a decline in curved nuclei as the number of cells in the tuft increased. Thus, the relative number of curved nuclei per tuft could also be used as a clear marker to distinguish medium and large tufts. As curved nuclei indicate cells are under severe mechanical strain, twisting or turning them around (*Xia et al., 2018*), this suggests that larger tufts may be more stable and mature, whereas the small and medium ones are under more tension, still forming with significant forces curving and pulling the cells around in the tuft. Interestingly, highly curved nuclei have been shown to result in rupture of the nucleus and DNA damage (*Xia et al., 2018*), which may further exacerbate dysfunctional cell behaviour in tuft formation. It should be noted that care should be taken to rotate the image stack to confirm nuclear curvature, as two nuclei parallel to each other can look like only one nucleus (*Figure 5a*, fourth panel row, blue arrow), emphasising the importance of 3D imaging with LSFM as rotating and viewing tufts from the side without distortion is not possible with confocal.

Finally, to quantify the level of distortion of vascular tufts incurred by flat-mounting and confocal imaging, we compared tuft depth measurements between retinas imaged with confocal and LSFM (depth defined the tuft length orientated perpendicular to plexus plane). The change in depth was particularly striking and more pronounced for larger tuft structures (*Figure 5a* bottom panels, e). Taken together, this further confirmed that LSFM is superior to confocal to image larger structures in the eye.

## LSFM OIR Case-study: Pathological retinal neovascular tufts have a swirling/knotted morphology

In order to gain better resolution to characterise the specific morphology of the different sized tufts we performed computational image deconvolution on cropped LSFM images of vascular tufts (see Materials and methods), which helped to decrease the scattered light caused by imaging thicker tissue with the light sheet without the need to clear the tissue (*Richardson and Lichtman, 2015*). Upon deconvolution a previously unappreciated 'knotted' morphology of the tufts was evident across all tuft classes; often tufts had one or more holes going through (*Figure 6a,b*; *Figure 6—figure supplement 1a,b* for more rotational views and original rotational *Videos 14,S15*). To describe these 3D tuft structures in detail, we first explored three systematic image analysis approaches: 1) by slowly shifting clipping planes through the tuft from the vitreous, facing side to the plexus-connecting side of the tuft, it was possible to better appreciate the upper and lower 3D organisation of the tuft; 2) carefully rotating and hand-drawing the tufts surface rendered structures from every angle and 3) comparing the colour-labelled positions of nuclei to indicate their depth position in the tuft. The first approach revealed that the tuft shown in *Figure 5a* fourth panel row, had a figure of eight knot, with two clear holes through the tuft and an unexpected vessel connecting the upper vitreous surface of the tuft to the plexus (*Figure 6c* and *Figure 2—figure supplement 1b*, *Video 13*). A combination of the second two approaches revealed a swirl structure to two tufts (small and medium in size), akin to a snake coiling upon itself in layers, with several highly curved nuclei (*Figure 6d–i*, *Videos 16* and *17*). We noted a central hole either through the entire tuft or evident in the upper vitreous facing portion, akin to a depression or invagination. A sprout-like protrusive tip with filopodia was often also evident (*Figure 7a*, *Figure 6—figure supplement 1c*). Overall, all three, 3D rotational image processing/analysis approaches were extremely useful for better interpreting these complex 3D structures, providing a much deeper understanding of tuft morphology than would be possible using a 2D analysis of distorted tufts, viewed from above in standard flat mount confocal microscopy.

To further validate these unexpected tuft morphologies with an independent high-resolution 3D imaging method, we performed microCT on intact health control and OIR retinas. We found microCT of mouse retinas entirely feasible and the OIR vascular tufts readily amenable to analysis by microCT, as they protrude into the vitreous (*Figure 6—figure supplement 2a*). On close inspection

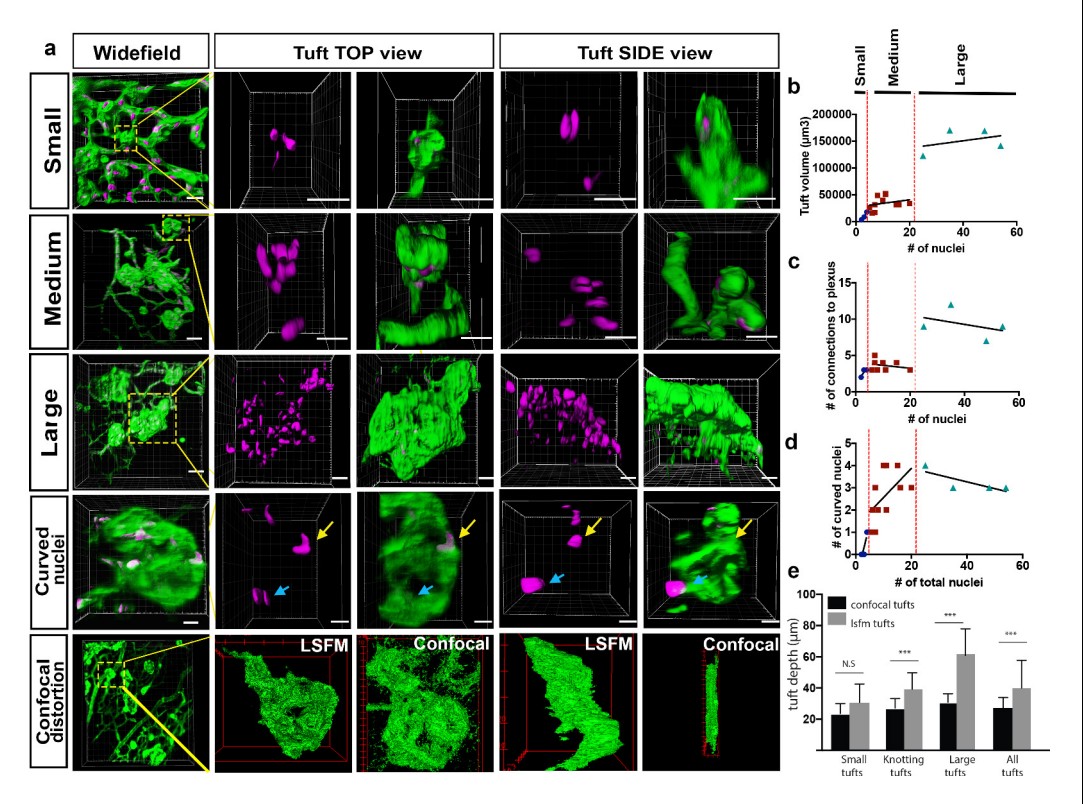

**Figure 5.** Analysis of three subclasses of OIR vascular tuft. (**a**) Representative 3D-rendered LSFM images of small (1st row), medium (2nd row), and large (3rd row) tufts showing the vasculature (IsoB4, green) and endothelial nuclei (ERG, magenta), scale bar = 10 µm. For all widefield images, scale bar = 40 µm, yellow box indicates tuft in situ; 4th row - representative 3D-rendered images showing curved nuclei in a medium tuft, yellow arrows indicate curved nuclei, blue arrows indicate flat nuclei parallel to each other. Scale bar, 10 um. 5th row: correlative LSFM-confocal microscopy of the same tuft reveals the tuft depth distortion (side view) incurred with confocal flatmounting versus LSFM. (**b**) The volume of the tufts versus the number of nuclei per tuft. (**c**) The number of vessel connections between the tuft and the underlying vascular plexus versus the number of nuclei. (**d**) Quantification of the number of curved nuclei per tuft versus the total number of total nuclei per tuft. (**e**) Quantifications of tuft depths per subclass size in LSFM vs confocal images, significant difference shown using unpaired t-test, *** means p<0.0001.

The online version of this article includes the following source data for figure 5:

**Source data 1.** Excel file containing source data pertaining to *Figure 5b–e*.

we indeed found tufts also appear to have holes/invaginations (*Figure 6—figure supplement 2b–c*) indicating further study of these complex 3D structures is warranted.

Next, we investigated whether LSFM could provide added benefits for OIR drug study quantifications, when compared to confocal microscopy. We therefore reproduced an OIR drug-treatment study using Everolimus, an inhibitor of the mammalian target of rapamycin (mTOR) and compared the feasibility of quantifications (primarily quantifying 2D avascular and/or tuft area/numbers) between LSFM and confocal microscopy (*Yagasaki et al., 2014*). In accordance to published data, we observed an evident increased avascular area and smaller tufts with drug treatment (*Figure 6—figure supplement 3a*). However, quantification of avascular area in LSFM was not feasible due to a lack of available computational tools to take account of the natural 3D curved retinal tissue surface, which suggests that confocal imaging is a more suitable imaging modality to quantify this 2D parameter. Yet, we found that LSFM was very practical to measure tuft volume and found that tufts in drug-treated retinas were markedly smaller than in untreated retinas, despite having comparable nuclear counts (*Figure 6—figure supplement 3b,c*). Furthermore, drug-treated retinal tufts showed a particular small swirl/ordered two-layer cup morphologies with distinctive wide-reading filopodia

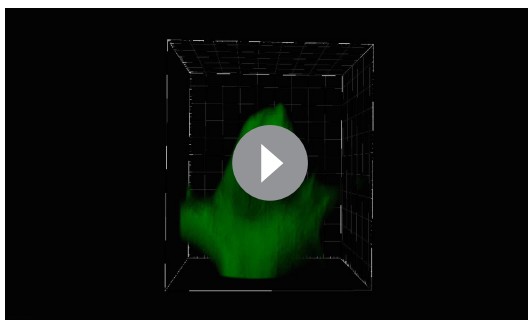

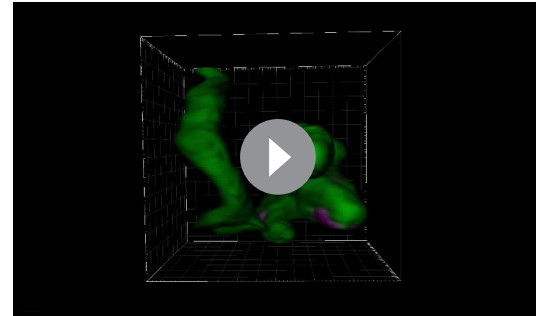

**Video 10.** 3D-rendered LSFM z-stack of a 'small tuft' from an OIR mouse retina stained for blood vessels (IsolectinB4, green), and the nuclear marker ERG (magenta). The z-stack was rendered in Imaris.
https://elifesciences.org/articles/49779#video10

**Video 11.** 3D-rendered LSFM z-stack of a 'medium tuft' from an OIR mouse retina stained for blood vessels (IsolectinB4, green), and the nuclear marker ERG (magenta). The z-stack was rendered in Imaris.
https://elifesciences.org/articles/49779#video11

all around the tuft, suggesting they are highly active (*Figure 6—figure supplement 3d–j*), similar to our previous observations in OIR untreated retinas (e.g. *Figure 6d*). Thus, we concluded that LSFM is more suitable for 3D volume and tuft morphology characterisation to understand the mechanism of action of OIR drug treatments than confocal microscopy.

## 4D LSFM live-imaging of the OIR mouse model reveals altered cell dynamics

To gain insights into endothelial cell behaviour in vascular tufts, we next imaged the OIR-induced tufts dynamically with LSFM. Thereby, we observed that filopodia extended/retracted from abnormal vascular tufts, similar to what is seen in the extending vascular front during development of the retina vasculature. However, filopodia formed from vascular tufts remained very short (mean 4.3 μm) as compared to normoxia (mean 14.84 μm) (*Figure 7a,b*). In OIR, filopodia more rapidly extended and retracted, without making connections (*Figure 7a,c,d*, *Video 18*). As the VEGF gradient is expected to be disrupted in the OIR model, timing from dissection to imaging is not as crucial. However, most filopodia movements occurred in the first few hours under this pathological condition. When imaging other parts of the OIR retinas to the tufts, we observed intriguing, abnormal EC behaviour. Their movements were undirected and appeared to involve blebbing-based motility (*Figure 7e,f*, *Video 19*). We observed both cells that were dividing, and undergoing apoptosis (*Figure 7e*, *Video 20*), which was not observed during normal conditions. This first live imaging of altered cell behaviour in the OIR mouse model further

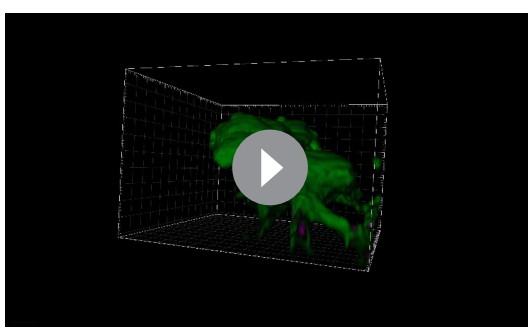

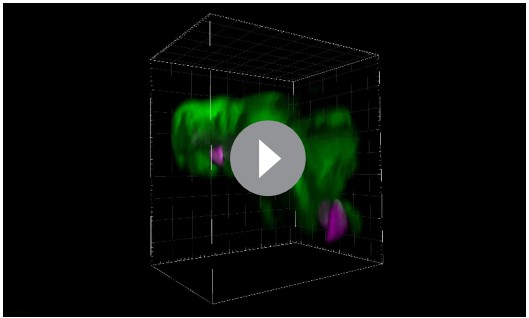

**Video 12.** 3D-rendered LSFM z-stack of a 'large tuft' from an OIR mouse retina stained for blood vessels (IsolectinB4, green), and the nuclear marker ERG (magenta). The z-stack was rendered in Imaris.
https://elifesciences.org/articles/49779#video12

**Video 13.** 3D-rendered z-stack showed a curved nuclei in a 'medium tuft' from an OIR mouse retina stained for blood vessels (IsolectinB4, green), and the nuclear marker ERG (magenta). The z-stack was rendered in Imaris.
https://elifesciences.org/articles/49779#video13

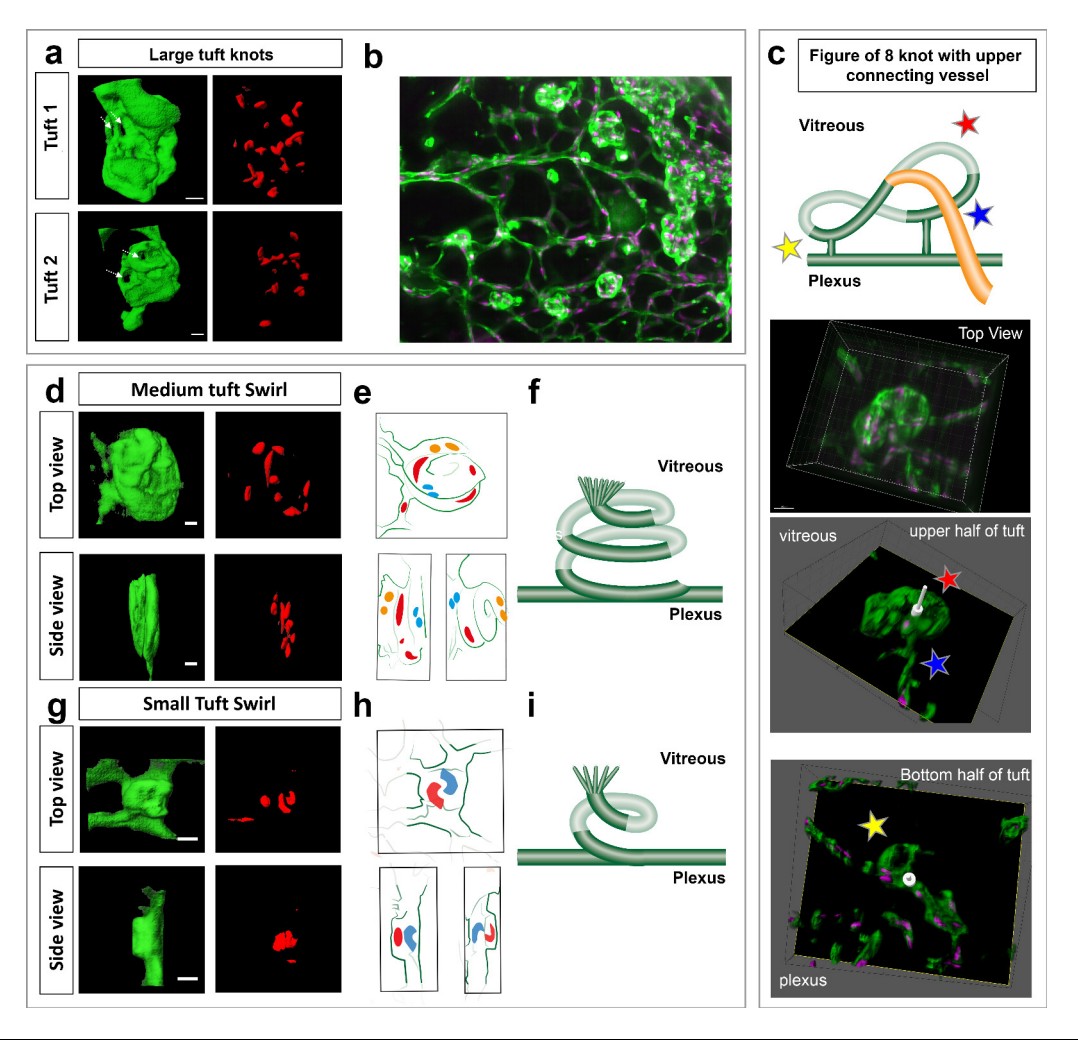

**Figure 6.** Knotted morphology of neovascular tufts revealed with LSFM. (**a,d,g**) Representative 3D-rendered images (generated using IMARIS software) of large (upper panel), medium and small tufts showing the vasculature (IsoB4, green) and endothelial nuclei (ERG, red) from rotational *Videos 14* and *15* See *Figure 6—figure supplement 1a* for further views from different angles of the large tufts. White dashed arrows indicates a hole through the tuft. Scale bar, 30 µm. (**b**) Widefield LSFM of OIR retina demonstrates that the knotted morphology is hard to discern from afar. (**c**) Detailed 3D clipping plane and 3D rotational drawings of an individual knot reveal a figure of eight structure with two clear holes through the tuft as well a vessel connecting from the upper, vitreous facing surface of the tuft to the plexus below (blue star). Stars mark corresponding regions from the illustration to the images - lower tuft loop nearer plexus (yellow star), upper tuft loop nearer vitreous (red star). See also *Figure 6—figure supplement 1b* for detailed 3D drawings made from each rotational view of this tuft with clipping planes, and *Video 13*. (**e,h**) 3D sketches made from rotational *Videos 16* and *17* to better elucidate nuclei: blue nuclei - bottom of tuft (near plexus), red nuclei – middle of tuft (in **e**), top of tuft (facing vitreous) in (**h**) yellow nuclei - top of tuft (facing vitreous) in **e**, (**f,i,**) schematic illustrating the swirling tuft morphology observed in (**d-h**) with three layers for the medium tuft (**f**) and two for the small one (**i**).

The online version of this article includes the following source data and figure supplement(s) for figure 6:

**Figure supplement 1.** LSFM reveals knotted morphologies in vascular tufts.

**Figure supplement 2.** MicroCT of OIR retinas shows tuft appearing to undergo invagination.

**Figure supplement 3.** Everolimus-treated tufts exhibit highly active filopodia and cup morphology.

**Figure supplement 3—source data 1.** Excel file containing source data pertaining to *Figure 6—figure supplement 3h and i*.

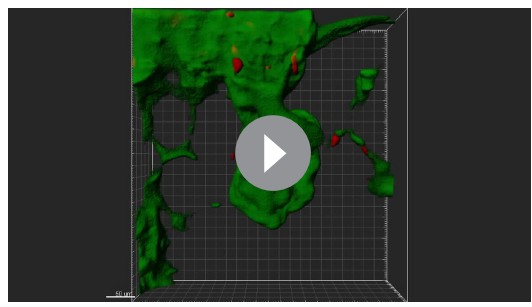 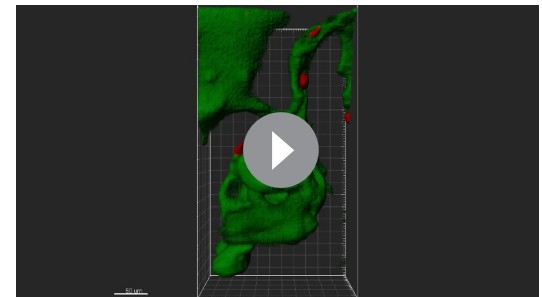

**Video 14.** 3D-rendered LSFM z-stack deconvolved in Huygens then reconstructed with surface rendering in Imaris of 'Large tuft 1' from an OIR mouse retina stained for blood vessels (IsolectinB4, green), and the nuclear marker ERG (red).
https://elifesciences.org/articles/49779#video14

**Video 15.** 3D-rendered LSFM z-stack deconvolved in Huygens then reconstructed with surface rendering in Imaris of 'Large tuft 2' from an OIR mouse retina stained for blood vessels (IsolectinB4, green), and the nuclear marker ERG (red).
https://elifesciences.org/articles/49779#video15

highlights the potential of LSFM for new insights into disease processes.

## Discussion

Although LSFM is becoming increasingly popular, studies to date have not attempted to use it to image mouse eye tissue. We have therefore established the first protocols to image and clear mouse eye tissue using LSFM. Because this protocol utilises optical sectioning of whole mount tissue, we found that LSFM is a very useful tool to rapidly image and reveal eye tissue at cellular and subcellular resolution without distortion of the sample due to flat-mounting, with the benefit to view, rotate and quantify structures in full 3D. As such, the present study provides a highly relevant and improved approach to examine the inter-relationships of normal neurovascular structures and the complex morphology of aberrant vascular structures in disease models, revealing for the first time an apparent knotted morphology to the vascular tufts in OIR. We have furthermore established an ex vivo 4D live-imaging method to follow angiogenic growth in the mouse retina in real-time, both during development and under pathological conditions, and feasibly quantified that these dynamics appear significantly altered in pathological conditions. The acquisition of 3D images of vascular structures at high spatial and temporal resolution within intact ocular tissue is both novel and significant. Overall, we strongly suggest the use of LSFM for 1) the study of larger or more complex 3D tissue structures reaching across the typical retinal layers, which are liable to distortion with standard approaches and 2) dynamic cell and subcellular processes in the mouse eye. *Singh et al. (2017)* established LSFM

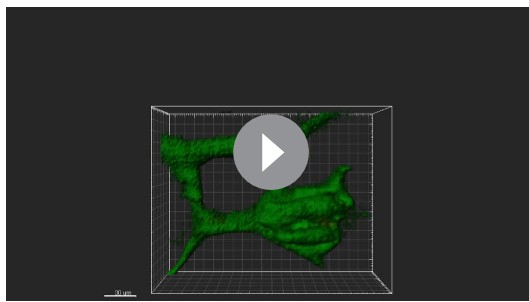 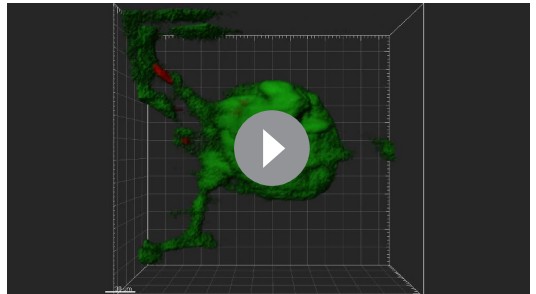

**Video 16.** 3D-rendered LSFM z-stack deconvolved in Huygens then reconstructed with surface rendering in Imaris of a 'small tuft' from an OIR mouse retina stained for blood vessels (IsolectinB4, green), and the nuclear marker ERG (red).
https://elifesciences.org/articles/49779#video16

**Video 17.** 3D-rendered LSFM z-stack deconvolved in Huygens then reconstructed with surface rendering in Imaris of 'a Medium swirl tuft' from an OIR mouse retina stained for blood vessels (IsolectinB4, green), and the nuclear marker ERG (red).
https://elifesciences.org/articles/49779#video17

imaging of rat retinas while this manuscript was in preparation, they focussed on static 3D imaging and analysis of vessels. Taken together with our results here in mice demonstrating LSFM for static 3D, live imaging and neuronal retinal studies in health and disease, this strongly indicates LSFM can bring improved 3D insights across rodent species for ocular development and disease studies. We see far-reaching potential of the approach for deeper insights into eye disease mouse models in particular. For example, it would now be feasible to skeletonise larger portions of the vascular network (ultimately, the entire retina vasculature), with accurate vessel morphometrics in order to perform flow simulations and understand how the biomechanical feedback of flow impacts vessel growth in healthy and diseased eyes.

## LSFM vs confocal: A balanced discussion

Benefits of LSFM:

*Speed -* in general image acquisition with LSFM is widely known to be far faster than confocal due to the illumination of the entire optical plane at once combined with the use of a camera instead of detectors, and an extensive stack of the entire mouse retina can be acquired very quickly using LSFM (~60 s). *Cost –* the instruments cost approximately the same, but as imaging is approximately ~10 x faster, the LSFM can be considered cheaper overall. *Depth -* LSFM is better for imaging thicker or very large tissues (such as the eye cup, which is thin, but topologically spherical), due to the fast acquisition rates and the large, rotatable sample holder, removing the limited single view point from above with upright microscopes and slide mounting. *Phototoxicity -* the illuminated plane generates less photobleaching and faster time frame rates for high temporal resolution live imaging of 3D/very thick tissues. We find LSFM imaging of the retina to be particularly informative over standard confocal microscopy when studying the following specific complex 3D and/or dynamic structures in the eye: 1) the adult retina in full - it is possible to visualise all three vascular layers in the LSFM, including direct cross-sectional viewing of the diving vessels oriented between layers by rotating the sample relative to the objective, which is not possible with confocal. Similarly, the iris and optic nerve can be observed in full, from any angle, undistorted with LSFM. 2) abnormally enlarged vessels/tufts - a new knotted morphological structure of tufts was apparent, and feasible to begin characterising due to the improved 3D imaging and rotational views possible with LSFM. With confocal imaging the tuft shape can only be inferred from above and we found the depths were significantly distorted and compressed, which is likely why knots have not been previously described. Interestingly, the VE-cadherin staining of endothelial junctions of several OIR tufts shown in *Bentley et al. (2014)* indicated there were 'holes' through tufts, as no junctional stains were found in clear pillars through them. However, the holes were not easy to confirm by isolectinB4 staining in those samples, likely due to spreading of the vascular structure when it was distorted during flat-mounting. We can confirm here with LSFM and microCT that holes and invaginations through tufts are evident and that tufts appear to consist of one or more long vessel structures intertwined, swirled and potentially looped upon themselves. 3) Neurovascular interactions in one sample, as neurons and vessels are oriented perpendicular to each other through the retina, they are normally imaged with separate physical sectioning or flattening techniques in either direction, prohibiting their concurrent observation. Optical sectioning of thick tissue and then rotating the undistorted image stacks allows both to be imaged together. Indeed, obtaining such images from one sample with LSFM will permit the quantification of vessels protruding through the neuronal layers, which is now only possible by performing time-consuming serial block-face scanning electron microscopy (*Denk and Horstmann, 2004*). 4) Subcellular level resolution in undistorted 3D retinal structures. We have shown that even in WT retinas, 3D analysis of subcellular structures such as the Golgi-nucleus polarity axis can be revealing, showing cells hidden beneath those that would be assumed as one using current 2D methods. However, we see the greatest potential for subcellular analysis in future studies analysing disruptions in cell polarity, or other processes at the subcellular level such as actin localisation in large pathological vessels or other retinal structures. 5) Live-imaging of developing mouse retinas – this has proven very difficult with in vitro methods providing more reliable assays, for example embryoid bodies (*Kearney and Bautch, 2003*; *Jakobsson et al., 2010*). Although embryoid bodies do form vessel-like structures, they are not perfused and do not fully reflect the complex and tissue specific in vivo retinal environment. Moreover, the embryoid bodies are treated with VEGF supplied to the culture medium, while in vivo, endothelial cells are exposed to a VEGF

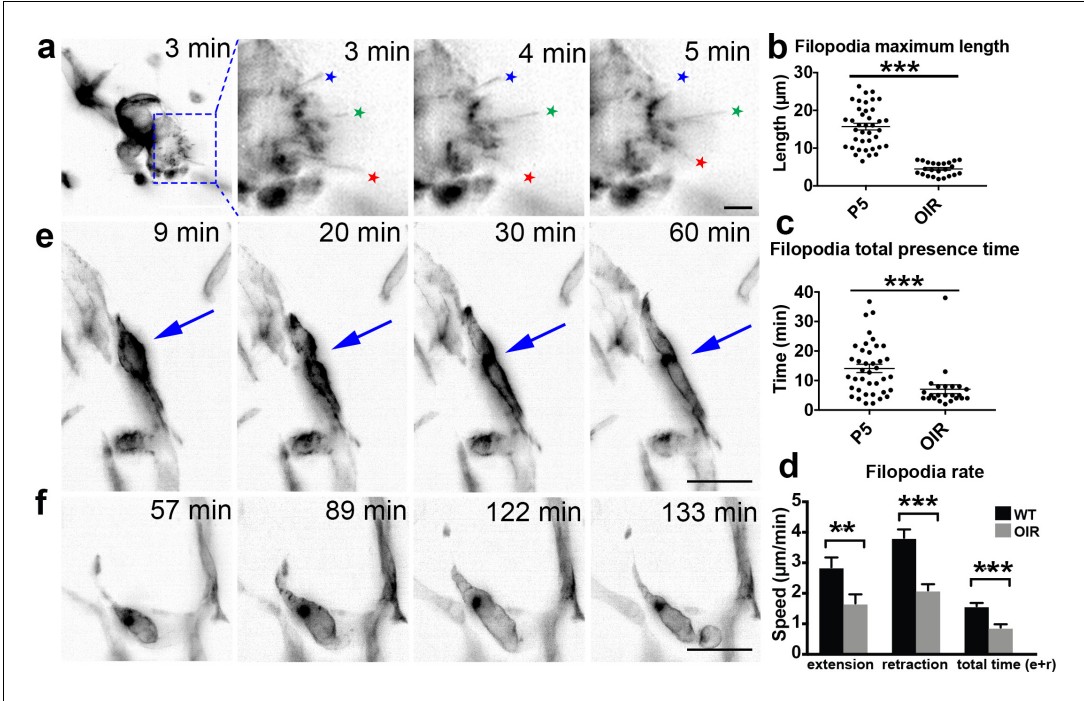

**Figure 7.** OIR live imaging. (a) Maximum intensity projections (MIPs) of a time lapse Video of a retinal tuft of a mouse in the oxygen-induced retinopathy (OIR) model visualise short, rapidly extending and retracting filopodia as compared to control retinas (stars). (b) The maximum length that each filopodia reached was measured for each filopodia over time in P5 and OIR conditions. (c) The total time that each filopodia was present during the experiment; this time is calculated from when one filopodia appeared and then disappeared. (d) Speed of extension and retraction of filopodia were calculated for P5 and OIR conditions. Total n = 67 and 23 filopodia in 8 and 3 cropped Videos from three independent P5 and 1 OIR experiment. (e) MIPs of a time lapse Video of mouse retinal vasculature in the OIR model reveal cell shuffling in real-time (arrow). (f) MIPs of a time lapse Video of mouse retinal vasculature in the OIR model reveal abnormal vessel growth in real-time. Scale bar, 20 µm. The online version of this article includes the following source data for figure 7:

**Source data 1.** Excel file containing source data pertaining to *Figure 7d–f*.

gradient from the astrocyte network below. Our images suggest that the VEGF gradient remains intact in the retina samples for several hours in LSFM imaging. Furthermore, our protocol enables us to follow and quantify filopodia movements from minute to minute, revealing movements never seen before. Thus, we observed astonishing abnormal cellular and subcellular level dynamics under pathological OIR conditions by 4D live LSFM imaging.

## Benefits of confocal over LSFM

Confocal microscopy has a fundamentally higher spatial resolution with less light scatter than LSFM; clearer, more precise images of smaller structures can be obtained, such as endothelial junctions and tip cell filopodia morphology, provided the tissue sample is amenable to flat mounting without distortion or loss of information – i.e. it is naturally thin cross-sectionally and structures of interest have their main features in the XY plane, not in Z, XZ or YZ. Thus, confocal static imaging of normal developing vessels in a single layer of the retina will still yield better resolution images than LSFM and is very reliable for XY based quantifications such as branch point analysis and 2D vascular area measurements in disease models/drug studies. However, we find it is not reliable for acquiring accurate quantifications involving depth through Z such as vessel diameters or the morphology of cells that span between the layers (e.g. in the XZ or YZ planes). Thus, overall LSFM is not suggested to replace confocal for static developmental angiogenesis studies or 2D analysis metrics on retinas. However, to study and measure precise morphological attributes or dynamics of vessels with

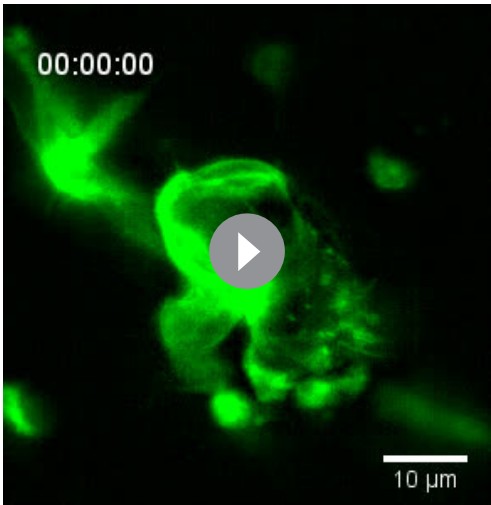

**Video 18.** A 2.5 hr time lapse Video of a retinal tuft of a mouse in the OIR model using mT/mG x cdh5 (PAC) CreERT2 mice visualise short, rapidly extending and retracting filopodia as compared to control retinas. Frame rate: one image/minute.
https://elifesciences.org/articles/49779#video18

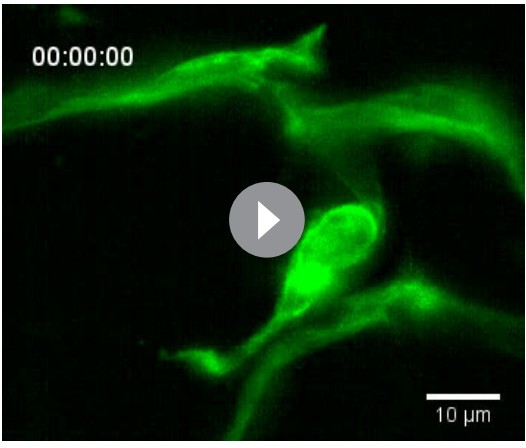

**Video 19.** A 2.5 hr time lapse Video of mouse retinal vasculature in the OIR model using mT/mG x cdh5 (PAC) CreERT2 mice reveal abnormal vessel growth in real-time. Frame rate: one image/minute.
https://elifesciences.org/articles/49779#video19

inherently 3D nature such as vessel radii, enlargements, malformations, diving vessels, iris, optic nerve or the deeper layers we find strong evidence to favor LSFM over confocal imaging.

In general, the quantification time was comparable between LSFM and confocal images but there is potential for image analysis to require more effort for LSFM as files can become quickly large due to the rapid imaging (~200 GB for static imaging and up to 4TB for live multichannel imaging). If an older eye is being imaged the three vascular layers will be somewhat visually overlapping (e.g. in *Figure 1—figure supplement 1b*), which could be hard to manually untangle due to the curvature, and as such represents a limitation. The preservation of the tissue depth information in the large Z-stack however, means by computationally fitting to the local curvature of the eye tissue one could computationally color code and subtract the retinal layers out for independent viewing and analysis, but this requires more investment than depth color coding of flat-mounted confocal images (*Milde et al., 2013*).

## Vascular tuft formation

The OIR model is a commonly used to study retinopathies. The three-dimensional nature of vascular tufts makes them ideal for LFSM and though this is a widely studied mouse model, the improved three-dimensional imaging allowed us to identify several new features of the important pathological vessels it generates. Our observations of small, medium and large tuft classes with distinct properties and the observation of more complex knotted, swirling and looping morphologies than previously reported, suggest a new mechanistic explanation is required to understand how and why vessels twist and turn on themselves and why it appears that medium tufts reach a critical size then stop twisting and instead coalesce into larger more stable structures, akin to the development of blood islands in retinal development (*Goldie et al., 2008*).

Nuclei with unusual shapes have previously been identified in abnormally growing tissues, such as cancer (*Hida et al., 2004*; *Kondoh et al., 2013*; *Versaevel et al., 2012*), and to reflect mitotic instability (*Gisselsson et al., 2001*). It is remarkable that we observed the dramatically curved shape of EC nuclei in tufts. Although it remains unclear whether their unusual shape has consequences for EC function in the tuft, it is tempting to speculate that it would have some bearing on, or is at least be an indicator of abnormal cell behavior. Overall, the ability to rotate the tufts in 3D and view from the side, not just the top, gave a much clearer view of their structure potentiating a detailed analysis of their complex knotted structure in the future. It was particularly interesting that tufts in the Everolimus-treated OIR retinas appeared to conform to a specific swirl structure with many filopodia,

suggesting that LSFM imaging could help reveal much greater information of the mechanism of action of many drugs targeting these or other complex 3D structures in the eye. LSFM therefore could greatly improve our understanding of these abnormal vascular formations, already opening up avenues for future studies.

### Reproducible live-imaging of angiogenesis in ex vivo retinas

Current retinal studies must infer dynamics from static images by hypothesising what might have happened in real-time to generate the retina's phenotype. For example, CollagenIV-positive and IsolectinB4-negative vessels are considered to be empty membrane sleeves where the vasculature has regressed. It is therefore important to establish reproducible live-imaging methods. It will be interesting to investigate in future live-imaging studies how pervasive the kiss and run behaviors are across the plexus and under different conditions, in order to fully elucidate their functional role. We furthermore demonstrated the potential to quantify diverse subcellular level movements in the cells and altered cell movements in the OIR disease model as proof of concept. Previously undirected vascular movements have been indicated as due to the loss of the underlying astrocyte template (*Dorrell et al., 2010*), LSFM now permits mechanisms involving multiple cell types to be investigated and confirmed live with fluorescent co-labelling studies of neurons/glial cells with vessels in the same retina. The LSFM live imaging protocol is sturdy as indicated from the testing in three different laboratories in three different countries (US, Sweden and Portugal) with different scientists performing the dissections and imaging, on different instruments. As such we can confirm that though challenging, the live imaging protocol has been optimised and is reproducible in different hands.

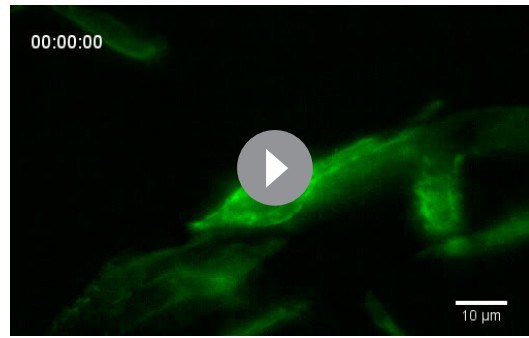

**Video 20.** A 2.5 hr time lapse Video of mouse retinal vasculature in the OIR model using mT/mG x cdh5 (PAC) CreERT2 mice reveal cell shuffling in real-time. Frame rate: one image/minute.
https://elifesciences.org/articles/49779#video20

## Materials and methods

**Key resources table**

| Reagent type (species) or resource | Designation | Source or reference | Identifiers | Additional information |
|---|---|---|---|---|
| Antibody | Anti-calretinin (Rabbit Polyclonal) | Abcam | Cat#:ab702, RRID: AB_305702 | (1:50) |
| Antibody | Anti-ERG (Rabbit Monoclonal) | Abcam | Cat#: ab92513, RRID: AB_2630401 | (1:200) |
| Antibody | Anti-calbindin (Rabbit Polyclonal) | Millipore | Cat#: AB1778 RRID: AB_2068336 | (1:300) |
| Antibody | Anti-GFAP (Rabbit Polyclonal) | Agilent, Dako | Cat#:Z0334 RRID: AB_10013382 | (1:100) |
| Antibody | Anti-collagenIV (Rabbit Polyclonal) | Bio-Rad | Cat#:2150–1470, RRID: AB_2082660 | (1:500) |
| Antibody | Neuron-specific beta-III Tubulin Biotin MAb (Clone TuJ-1) (Mouse monoclonal) | R and D Systems | Cat#: BAM1195, RRID: AB_356859 | (1:50) |
| Antibody | Anti-Smooth Muscle Actin (Mouse monoclonal) | Sigma-Aldrich | Cat#: C6198, RRID: AB_476856 | (1:1000) |

*Continued on next page*

*Continued*

| Reagent type (species) or resource | Designation | Source or reference | Identifiers | Additional information |
|---|---|---|---|---|
| Antibody | Anti-MGOLPH4 (Rabbit Polyclonal) | Abcam | Cat#: ab28049 RRID: AB_732692 | |
| antibody | Anti-Mouse CD31/PECAM-1 (Goat polyclonal) | R and D System | Cat#: AF3628 RRID: AB_2161028 | (1:200) |
| Commercial assay or kit | CyGEL | biostatus | CAT#: CY10500 | (1:400) |
| Software, algorithm | Arivis Vision 4D | arivis AG | N/A – (A new RRID has been requested as of paper submission). | |
| Software, algorithm | Imaris | Bitplane | RRID:SCR_007370 | |
| Software, algorithm | FIJI | Shcindelin, 2012 | RRID:SCR_002285 | StackReg plugin used |
| Software, algorithm | Zeiss Scout and Scan Control Reconstructor Software | Zeiss | N/A | |
| Software, algorithm | Drishti | *Limaye, 2012* | N/A – (A new RRID has been requested as of paper submission). | |
| Software, algorithm | Zeiss Zen | Zeiss | RRID:SCR_013672 | |
| Software, algorithm | Huygens Software | Scientific Volume Imaging | RRID:SCR_014237 | |
| Other | Alexa Fluor 568 isolectin GS-IB4 conjugate | Thermo Fisher Scientific | Cat#:I21412 | Molecular probe. |
| Other | Alexa Fluor 488 isolectin GS-IB4 conjugate | Thermo Fisher Scientifc | Cat#:I21411 | Molecular probe. |
| Other | Alexa Fluor 647 isolectin GS-IB4 conjugate | Thermo Fisher Scientifc | Cat#: I32450 | Molecular probe. |
| Other | Draq5 | Thermo Scientific | Cat#: 62251 | Cell (DNA) stain |

## Mice

mT/mG mice (*Wang et al., 2010*) were crossed with cdh5 (PAC) CreERT2 mice. For live-imaging of retinal angiogenesis during development, mice were injected with 50 µg tamoxifen at postnatal day (P) 1, P2 and P3, and imaged at P4 (*Wang et al., 2010*). For live-imaging of oxygen-induced retinopathy (OIR) experiments, mice were injected with 100 µg tamoxifen at P13, P14 and P15. The retinal vasculature was imaged at P17 unless otherwise stated. Recombination was confirmed by GFP expression in ECs. GNrep mice (*Barbacena et al., 2019*) were injected with 4OH-tamoxifen at P3 and P4 (20 ug/g) and fixed in PFA 2%. Lifeact mice (*Riedl et al., 2010*) were a kind gift from Dr. Wedlich-Söldner, University of Münster, Germany. Mice used in experiments at Beth Israel Deaconess Medical Center were held in accordance with Beth Israel Deaconess Medical Center IACUC guidelines (protocol #009–2014). Animal work performed at Uppsala University was approved by the Uppsala University board of animal experimentation (ethics approval reference C134/14 and C116/15). Animal work performed at SERI was IACUC approved (protocol S467-1019). Animal work performed at FAS Harvard was IACUC approved (protocol 14-02-191). Transgenic mice were maintained at the Instituto de Medicina Molecular (iMM) under standard husbandry conditions and under national regulations (DGAV project license 0421/000/000/2016.

## Antibodies

IsolectinB4 directly conjugated to Alexa488 and all corresponding secondary alexa conjugated antibodies were obtained from Invitrogen. Isolectin IB4 conjugated with an Alexa Fluor 568 dye was purchased from Thermo Fisher Scientific, MA. Anti-calretinin (ab702) and anti-ERG (ab2513) antibodies were obtained from Abcam. The antibody directed against Calbindin (AB1778) was acquired from Millipore. Anti-Glial Fibrillary Acidic Protein (GFAP) antibody was purchased from Dako (Z0334), anti-CollagenIV from AbD Serotec (2150–1470), biotinylated anti-neuron-specific b-III Tubulin from R and D Systems (Clone TuJ-1, BAM1195), and Cy3-conjugated anti-smooth muscle actin (SMA) antibody was obtained from Sigma Life Science (C6198). Draq5 was obtained from ThermoScientific. Anti-GOLPH4 (ab28049) from Abcam. GNrep mice were co-stained with CD31 (R and D, AF3628, 1/200) and anti-RFP antibody coupled to mCherry (Alfagene, M11240, 1/100) to further increase the signal. TO-PRO-3 stain for Rho KO neurodegeneration study (Thermo Fisher Scientific; diluted 3000x in PBS).

## Immunohistochemistry

Retinas were dissected as previously described (*del Toro et al., 2010*). In brief, eyeballs were fixed for 18 min in 4% paraformaldehyde at room temperature. After dissection, retinas were blocked for 1 hr in blocking buffer (TNBT) or Claudio's Blocking Buffer (CBB) for retinas with Golgi stained. CBB consists of 1% FBS (ThermoFisher Scientific), 3% BSA (Nzytech), 0.5% Triton X100 (Sigma), 0.01% Sodium deoxycholate (Sigma), 0,02% Sodium Azide (Sigma) in PBS pH = 7.4 for 2 hr in a rocking platform) for retinas stained for golgi. Thereafter, retinas were incubated overnight in primary antibody in blocking buffer. After extensive washing, retinas were incubated in the corresponding secondary antibody for 2 hr at room temperature. For confocal microscopy, retinas were mounted on glass slides, and for LSFM, retinas were mounted in 2% low-melting agarose. Agarose was melted at >65 °C, and then maintained at 42 °C before adding the tissue. To minimise curling of the retina, apply 1–2 drops of low melting agarose on retina and start uncurling the retina before the gel is solidified. It can then be transferred to the cylinder for imaging.

## PACT clearing of retinas

PACT clearing was performed as previously described (*Treweek et al., 2015*). Retinas were dissected and fixed with 4% PFA at 4 °C overnight. Samples were incubated overnight at 4 °C in ice cold A4P0 (40% acrylamide, Photoinitiator in PBS). The following day, samples were degassed on ice by applying a vacuum to the tube for 30 min, followed by purging with $N_2$ for 30 min. Thereafter, samples were incubated at 37 °C for 3 hr to allow hydrogel polymerisation. Excess gel was then removed from the samples, the samples washed in PBS, and incubated at 37 °C for 6 hr in 8% SDS/PBS, pH 7.5. Samples were then washed in PBST for 1–2 days, changing wash buffer 4–5 times to remove all of the SDS. Immunostaining was then performed following the same protocol without PACT clearing. Thereafter, the tissue was cleared by at least 48 hr incubation in RIMS (40 g histodenz in 30 ml of sterile-filtered 0.02 M phosphate buffer, 0.01% sodium azide). Cleared retinas were mounted in 5% low-melting agarose/RIMS for LSFM imaging.

## Neuronal degeneration study on Rho KO retinas

Wild-type control and Rho knockout eye cups were incubated overnight in TO-PRO-3 stain (Thermo Fisher Scientific; diluted 3000x in PBS). Rho KO eye cups were then washed thoroughly with PBS for 24 hr prior to clearing with a modified iDISCO+ protocol (*Renier et al., 2016*). Briefly, eye cups were dehydrated through a methanol/water gradient (20%, 40%, 60%, 80%, 100%, 100%). Incubations were for 30 min at each concentration. Next, eye cups were incubated twice in 100% dichloromethane for 30 min. Finally, eye cups were transferred to 100% ethyl cinnamate and incubated for at least 1 hr prior to imaging. Eye cups were imaged in ethyl cinnamate using a Lightsheet microscope (Zeiss, Jena Germany) with modified optics designed for imaging 1.56 refractive index solutions. A 20 × 1.0 NA objective (RI = 1.56 corrected) was used for detection.

## Live-imaging

For live-imaging, mice were imaged at P4 or P5. The sample chamber was filled with DMEM without phenol red containing 50% FBS and P/S and heated to 37 °C. Retinas were quickly dissected in

prewarmed HBSS containing penicillin and streptomycin. After dissection, retinas were rapidly cut into quarters (mainly to minimise the datafile size created, the curved form was preserved) and immediately mounted in 1% low melting agarose in DMEM without phenol red containing 50% Fetal Bovine Serum (FBS) and 1x penicillin and streptomycin (P/S). To minimise curling of the retina, as with static imaging, apply 1–2 drops of low melting agarose on retina and start uncurling the retina before the gel is solidified. It can then be transferred to the cylinder for imaging.

## MicroCT

Eyes were dissected and fixed for one hour in 4% PFA in 0.1 M PB pH 7.4. The retinas were isolated, and post-fixed overnight in 4% PFA plus 2.5% GA in 0.1 M PB pH 7.4 prior to storage in 1% PFA in 0.1 M PB. After washing in 0.1 M PB to remove fixative residues, secondary fixation was performed in 2% reduced osmium tetroxide (aqueous), followed by washes in H2O and storage at 4°C. For hydrated microCT imaging, individual retinas were mounted in CyGEL (Biostatus, Shepshed UK), and scanned using an Xradia 510 Versa (Zeiss). Scans were performed at 40 kV/3 W using an exposure of 10 or 20 s and 3001 projections (OIR overviews, 1.89 µm voxels,~3 mm field of view; WT overviews 3.78 µm voxels,~3 mm field of view). The data was reconstructed into 16-bit TIFF image sequences using Scout-and-Scan Control System Reconstructor software (Zeiss). For visualisation of retina over-views, the OIR datasets were binned in XYZ to reach a voxel resolution of 3.78 µm, thereby matching the WT datasets, and rendered in three dimensions using Drishti (*Limaye, 2012*). To visualise individual epiretinal tufts, the full resolution OIR datasets were cropped to smaller regions of interest, and rendered in three dimensions using the volume viewer plugin in Fiji, and Imaris.

## Everolimus drug-treatment

Pups (P7) were put into the OIR chamber with the dam and exposed to 75% of O2 during 5 days (P11). At P11 animals were passed to normoxia conditions and injected with Everolimus (P11-P12-P13-P14) during four consecutive days. Sacrificed at P15 and retinas collected. Eyes were fixed with 2% of PFA for 5 hr. Everolimus (Selleckchem) treatment administered with subcutaneous injections of 5 ug/g of Everolimus and the Vehicle (DMSO + 30% PEG300).

## LSFM equipment

All LSFM images were acquired with a Zeiss Z.1 light sheet microscope except for those detailed below. The Zeiss objectives used for uncleared tissue and live imaging were Zeiss, RI = 1.33, 5x/0.16, and 20x/1.0. For PACT cleared tissue, RI = 1.45, 20x/1.0 (5.5mm working distance) was used. The Luxendo-Bruker MuVi-SPIM was used for specialised subcellular imaging of Golgi (*Figure 1—figure supplement 1e–h*) and tuft morphology in *Figure 6—figure supplement 3d–j*. The Miltenyi-LaVision BioTec Ultramicroscope II light sheet microscopes is particularly good for larger samples and used for the overview images in *Figure 6—figure supplement 3a*. The Luxendo objectives used were Olympus, RI = 1.33, 20x/1.0 in combination with a 1.5x magnification changer. The LaVision objective used was a Olympus MV PLAPO 2XC/0.5 in combination with a 2x zoom.All raw data were handled on a high-end DELL workstation (Dual 8-core Xeon Processors, 196 GB RAM, NVIDIA Titan Black GPU, Windows 7 64 bit) running ZEISS ZEN (Light sheet edition) or equivalent. Confocal images were taken with the LSM 880 Confocal Microscope.

## Image analysis

### Visualisation of images in 3d

3D reconstructions of images up to 4 GB were obtained using Imaris software. Fiji was used for reconstruction of images larger than 4 GB. To quantify tuft volumes, Arivis Vision4D software was used.

### Visualisation of live-imaging

To visualise live-images, the maximum intensity projection of each timepoint was made in ZEN (Zeiss). The Videos were corrected for drift correction in Fiji using the StackReg plugin and the background subtracted in Fiji using rolling ball background subtraction.

## Deconvolution

In LSFM, when using a high NA (>~0.6) objective, the optical section is determined by the depth of field of the objective and not the light sheet. However, in our system the light sheet is thicker than the objective's depth of field and substantial out-of-focus light is captured relative to confocal. Additionally, thick tissue samples have an intrinsic milky appearance. This lack of clarity undermines sharp images and becomes progressively more of an impediment the deeper one tries to look into a tissue volume. This translucency is caused by heterogeneous light scattering (*Richardson and Lichtman, 2015*). As the tissue used for imaging in LSFM is thick, fluorescent light originating from deep within the tissue is scattered during its travel through the tissue volume, back to the objective. This results in both in- and out-of-focus light arriving at an incorrect position on the camera causing objects to blur.

To deconvolve and reduce this light scatter computationally images were split into channels with their respective emission wavelength. Microscopic parameters (including pixel size, objective and excitation wavelength) were inserted into the settings of Huygens software for each channel followed by choosing the signal to noise ratio (SNR) for each image and run the deconvolution with the same settings. The resulted deconvolved images were inserted into Imaris for further analysis in 3D if needed. Huygens software was used for deconvolving all images.

## Actin-rich bundles and filopodia tracking

Tracking was performed manually using ImageJ/Fiji. For filopodia tracking, each filopodia was tracked between each frame of imaging and different analysis was performed. For tracking the actin-rich bundles, the Manual Tracking plugin in Fiji was used to manually select the ROI (=region of interest) and follow the pathway of each trajectory. The trajectories and the pathway were overlaid. Each trajectory could be visualised using Montage function.

## Neuronal degeneration rho KO mouse retinal analysis

Nuclei Density Method: Using freehand selection in FIJI to define the ONL area in a given image slice, we performed a particle analysis after thresholding to obtain an estimate of nuclei density (n = 1 retina per condition). This was performed at multiple points (n = approx. eight areas per condition) in the retina by incrementing the Z-dimension 50 slices and retaking the measurements, before averaging across the densities across slices. ONL Thickness Method: To measure average ONL thickness (n = 1 retina per condition), we used straight line selection in FIJI at three points along the ONL in a given slice and averaged the lengths. This was performed at multiple points (n = approx. eight slices per condition) in the retina by incrementing the Z-dimension 50 slices, before retaking the measurements and averaging across slices to obtain an estimate for the overall ONL thickness.

## Acknowledgements

We would like to thank Sven Terclavers (HCBI) for the excellent technical support; Alessandro Ciccarelli (The Francis Crick Institute, CALM STP) for LSFM imaging using the Luxendo and LaVision LSFM microscopes; Christopher Peddie and Lucy Collinson at The Francis Crick Institute (EM STP) for establishing and performing retinal microCT imaging; thanks also to Joe Brock and the Illustration team at The Francis Crick Institute for aiding with 3D drawing of tuft knots. CP and KB were supported by funding from Harvard Catalyst | The Harvard Clinical and Translational Science Center (National Center for Research Resources and the National Center for Advancing Translational Sciences, National Institutes of Health Award UL1 TR001102), the NEI (1R21EY027067-01), and BIDMC. K B and PA were supported by The Kjell and Märta Beijer Foundation. PA was additionally supported by a travel grant from The Royal Swedish Academy of Sciences (Kungl. Vetenskaps-Akademien). KB and LCW were supported by a grant from the Knut and Alice Wallenberg foundation (KAW 2015.0030). KB and TM were supported by The Francis Crick Institute, which receives its core funding from Cancer Research UK (FC001751), the UK Medical Research Council (FC001751), and the Wellcome Trust (FC001751). LV was supported by a Victor A McKusick fellowship from the Marfan Society. MR was supported by an EMBO fellowship (ALTF 2016–923). KIH was supported by institutional training grant T32 HL07893 from the NHLBI of the NIH. LV was funded by the Victor A

McKusick Fellowship from the Marfan Foundation and BIDMC. MO funded by European Union's Horizon 2020 Marie Skłodowska-Curie actions (842498). DFC supported by EY025259, Lions Foundation, and NEI core grant P30 EY03790. K-S Cho: EY027067. CAF was supported by European Research Council starting grant (679368), the Fundação para a Ciência e a Tecnologia funding (grants: IF/00412/2012; PRECISE-LISBOA-01–0145-FEDER-016394; and a grant from the Fondation Leducq (17CVD03).

## Additional information

### Funding

| Funder | Grant reference number | Author |
| --- | --- | --- |
| National Eye Institute | 1R21EY027067-01 | Claudia Prahst<br>Katie Bentley |
| Harvard Catalyst | UL1 TR001102 | Claudia Prahst<br>Katie Bentley |
| Beth Israel Deaconess Medical Center | startup funds | Claudia Prahst<br>Lakshmi Venkaraman<br>Katie Bentley |
| Kjell och Märta Beijers Stiftelse | | Parham Ashrafzadeh<br>Katie Bentley |
| Marfan Foundation | Victor A McKusick fellowship | Lakshmi Venkaraman |
| European Molecular Biology Organization | ALTF 2016-923 fellowship | Mark Richards |
| National Heart, Lung, and Blood Institute | T32 HL07893 | Kyle Harrington |
| National Eye Institute | EY025259 | Dong Feng Chen |
| National Eye Institute | P30 EY03790 | Dong Feng Chen |
| European Research Council | starting grant (679368) | Claudio A Franco |
| Fundação para a Ciência e a Tecnologia | IF/00412/2012 | Claudio A Franco |
| Fondation Leducq | 17CVD03 | Claudio A Franco |
| National Eye Institute | EY027067 | Kin-Sang Cho |
| Knut och Alice Wallenbergs Stiftelse | KAW 2015.0030 | Lena Claesson-Welsh<br>Katie Bentley |
| Francis Crick Institute | | Thomas Mead<br>Katie Bentley |
| Fundação para a Ciência e a Tecnologia | PRECISE-LISBOA-01-0145-FEDER-016394 | Claudio A Franco |
| Royal Swedish Academy of Sciences | | Parham Ashrafzadeh |
| H2020 Marie Skłodowska-Curie Actions | 842498 | Marie Ouarné |

The funders had no role in study design, data collection and interpretation, or the decision to submit the work for publication.

### Author contributions

Claudia Prahst, Conceptualization, Data curation, Formal analysis, Supervision, Funding acquisition, Validation, Investigation, Visualization, Methodology, Project administration; Parham Ashrafzadeh, Conceptualization, Data curation, Formal analysis, Supervision, Funding acquisition, Validation, Investigation, Visualization, Methodology; Thomas Mead, Formal analysis, Validation, Investigation, Visualization, Methodology; Ana Figueiredo, Software, Formal analysis, Investigation, Visualization,

Methodology; Karen Chang, Lakshmi Venkaraman, Ana Martins Russo, Formal analysis, Validation, Investigation; Douglas Richardson, Resources, Formal analysis, Supervision, Investigation, Methodology; Mark Richards, Kin-Sang Cho, Resources, Formal analysis, Supervision, Investigation; Kyle Harrington, Software, Formal analysis, Validation, Visualization, Methodology; Marie Ouarné, Resources, Supervision, Validation, Investigation, Methodology; Andreia Pena, Resources, Supervision, Investigation, Methodology; Dong Feng Chen, Resources, Supervision, Funding acquisition, Validation, Investigation, Methodology; Lena Claesson-Welsh, Resources, Supervision, Funding acquisition, Validation, Investigation; Claudio A Franco, Resources, Supervision, Validation; Katie Bentley, Conceptualization, Supervision, Funding acquisition, Investigation, Visualization, Methodology, Project administration

## Author ORCIDs
Thomas Mead (iD) https://orcid.org/0000-0003-2728-670X
Dong Feng Chen (iD) http://orcid.org/0000-0001-6283-8843
Lena Claesson-Welsh (iD) http://orcid.org/0000-0003-4275-2000
Kin-Sang Cho (iD) http://orcid.org/0000-0003-4285-615X
Claudio A Franco (iD) http://orcid.org/0000-0002-2861-3883
Katie Bentley (iD) https://orcid.org/0000-0002-9391-659X

## Ethics
Animal experimentation: This study was performed in strict accordance with the recommendations in the Guide for the Care and Use of Laboratory Animals of the National Institutes of Health. Mice used in experiments at Beth Israel Deaconess Medical Center were held in accordance with Beth Israel Deaconess Medical Center institutional animal care and use committee (IACUC) guidelines. Animal work performed at Uppsala University was approved by the Uppsala University board of animal experimentation. Transgenic mice were maintained at the Instituto de Medicina Molecular (iMM) under standard husbandry conditions and under national regulations.(ethics approval reference C134/14 and C116/15).

## Decision letter and Author response
Decision letter https://doi.org/10.7554/eLife.49779.sa1
Author response https://doi.org/10.7554/eLife.49779.sa2

## Additional files
### Supplementary files
• Transparent reporting form

### Data availability
All data generated or analysed during this study are included in the manuscript and supporting files. Data has been provided for Figures 3d, e, Figure 4c, Figures 5b,c,d,e, Figures 7d,e,f, Supp. Figures 2c,d and Supp. Figures 5h, i.

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
