## [Decision Letter]

**Acceptance summary:**

In this paper, the authors describe novel use of light sheet fluorescence microscopy for imaging and analysis of cellular and tissue morphology of mouse retinas during development and disease. The improvements described for ex vivo live imaging, as well as co-staining and analysis of neuronal and vascular systems together are particularly impressive and represent an important methodological advance in the field.

**Decision letter after peer review:**

[Editors’ note: the authors submitted for reconsideration following the decision after peer review. What follows is the decision letter after the first round of review.]

Thank you for submitting your work entitled "3D/4D characterization of cell behavior in the mouse retina in health and disease" for consideration by *eLife*. Your article has been reviewed by three peer reviewers, and the evaluation has been overseen by a Senior Editor. The following individuals involved in review of your submission have agreed to reveal their identity: Luisa Iruela-Arispe (Reviewer #2).

Our decision has been reached after consultation between the reviewers. Based on these discussions and the individual reviews below, we regret to inform you that your work will not be considered further for publication in *eLife*. Although the reviewers found aspects of your study, particularly the methodology, to be of high quality the overall assessment was that the study was not sufficiently well developed for eventual publication in *eLife*.

*Reviewer #1:*

In their manuscript "3D/4D characterization of cellular behavior in the mouse retina in health and disease" Prahst and colleagues demonstrate the potential of Light Sheet Fluorescence Microscopy (LSFM) in imaging complex tissues such as the mouse retina in 3D and 4D.

In the first part the authors describe the establishment of this imaging technique, which they then directly and quantitatively compare to images obtained by standard confocal microscopy on fixed material. They further show that LSFM is a powerful technique to investigate retinopathies such as in the OIR model. Importantly, LSFM can applied to live imaging of retinas in ex-vivo preparations.

Overall, this work is of high quality and demonstrates that the application of LSFM microscopy will be a useful technique to study angiogenesis especially when improved 3D resolution is warranted. The potential for LSFM in live imaging appears to be limited in spatial resolution and time.

At this point I have three concerns.

1) In Figure 3 the authors perform comparative analyses of a sample image acquired by LSFM and confocal microscopy. The confocal image (Figure 3A) appears to be saturated, which makes quantitative analysis difficult. In general, the authors should provide histograms to ensure that comparable images are being analyzed. Also, the pixel depth of images should be specified. Generally, most of the data presented are rendered images. A better representation of less processed images would be helpful.

2) When considering alternatives to confocal imaging, e.g. live imaging. The authors should also discuss optical coherence tomography (OCT), which has also been successfully applied for in vivo imaging of the mouse retina.

3) To assess the spatial resolution of LSFM compared to other imaging techniques it would be helpful to examine finer morphological structures – for example cell junction, Golgi or mitochondria.

*Reviewer #2:*

The present manuscript communicates technology that enables visualization of the vasculature of the retina in the context of its 3D structure using light sheet microscopy. The authors argue that this technology is superior than observing the retina as a flatmounted structure. Basically, this is a methodological paper that provides a new approach on how to visualize angiogenesis, vascular dynamics and vascular pathological processes in a manner that respects the geometry of the eye.

Technical advancements go hand-and-hand with scientific progress, one pushing the other, thus I am happy to see that *eLife* submits to review cutting-edge methodologies that could significantly impact specific fields. A critical aspect of this manuscript is the value-added of this new technology. As an outside evaluator, I see both positives and negatives of this technology and it would be critical for the authors to provide a more balanced view of these. Or perhaps, if I am missing something, explain to the review team what is the misinterpretation on the aspects of the study indicated below.

1) A detailed evaluation of the images provided reveals the lack of a clear hierarchy when retinas are evaluated by LSFM versus confocal flat-mounted images. For example, Figure 1F shows the vasculature of a P10. At this time, the vessels should clearly depict hierarchic branching patterns with alternating arteries and veins. Instead, the image shows a mesh or vessels that lack hierarchical organization and absence of alternating arterial-vein vessels. Is this outcome due to the fact that three vascular layers are juxtaposed, providing excessive noise to the inner layer (the hierarchical layer)? If this is the case, there are several negatives associated with this approach, although again, there are several positives (important to bring balance to the manuscript).

2) For the visualization shown in Figure 3: Was this done using a flat mounted image? Or Having the eyeball as a curved structure? I believe it might be the former, and in this case, it is difficult to understand why there are differences in the evaluation. The authors offer that assessment of the vasculature using lighsheet shows wider vessels than confocal, but the problem with the out-of-focus light discussed by the authors brings a confounding factor on the metrics. Can the authors discuss this at greater length? Also how do these metrics compared to a perfusion-fixed transversal section of specific vessels?

3) It is hard to interpret Figure 4 without a full legend (missing in the manuscript) but the comparison between confocal and LSFM clearly shows the advantage of LSFM, as the tufts are not squeezed through mounting. However, I am having a hard time in really understanding what is a "mature" versus a "knotting" tuft. If a nomenclature is to be introduced, then a more detailed description, assessment of multiple retinas, etc should be provided.

4) Clearly a big advantage of LSFM is the live imaging shown in Figure 5. Because there are multiple image modalities, it will be important to clearly state how is the ex-vivo preparation (flat? Curved? With media? Without? Heat? Oxygen?).

*Reviewer #3:*

Prahst and co-authors report the advantages of using light-sheet fluorescence microscopy (LSFM) to image the mouse retina in its natural spherical conformation. The mouse eye is of particular interest to ophthalmology, neurosciences and vascular biology. Thus, methods improving the capacity to characterize the biological and cellular principles governing the development and function of the mouse eye will have a broad-spectrum impact.

Authors start by claiming that visualizing blood vessels in the retina's natural spherical conformation will be important for our understanding of vascular biology. Thus, a central point is to compare this new method with already existing methods, such as confocal microscopy. The only comparison done between both is related to vessel diameter, and no changes were found. Other quantitative information used in vascular biology includes: vascular density; number of branching points; number of endothelial tip cells; number of filopodia per tip cell; number of regressing branches; etc. Does LSFM improve the analysis of those parameters? It is likely not the case. Moreover, description of the deconvolution and segmentation parameters in Materials and methods is very vague and relies heavily on "experts" (subsections “Manual cropping of well-segmented regions” and “Isosurface extraction”).

Another advantage of LSFM could be that the new method is faster when coming to perform the same task (extract quantitative information of fluorescence images). However, authors did not discuss this aspect. It would be important therefore to include the amount of time needed to image and quantify these parameters per retina and compared it to confocal microscopy.

Authors also showed the capacity to perform live imaging of endothelial cells in the mouse retina. Despite being technically possible, the reviewer is sceptical on the value of the information that can be obtained in this way. Given the tremendous change in the environment, including oxygen concentrations, loss of vitreous humor, lack of blood flow, loss of intraocular pressure, which are known to affect endothelial cell biology, it is debatable how reliable can be any information acquired with this protocol. As a matter of fact, Sawamiphak et al., 2010 reported a similar protocol, but the impact in the field so far is very minor. Moreover, no quantification of cell migration or filopodia dynamics was attempted in order to illustrate the potential value of the method. The same holds true in regard to the pathological OIR model.

A potential interest of the report comes from the analysis of vascular malformations, so-called vascular tufts, in the OIR model. Authors describe in high detail different vascular structures including tufts of different sizes, ranging from 2 to >20 endothelial nuclei, including the presence of some curved nuclei. Despite the novelty aspect of the observations, the reviewer is not convinced that the same structures could not be observed in confocal microscopy following careful analysis. Within the same context, authors claimed "abnormal cell dynamics" in the context of pathological angiogenesis. However, the manuscript includes very limited data (one video on filopodia, and 2 videos on undefined vessels) and no quantitative information to back up authors' claims. They suggest that those observations confirm previous hypotheses (Bentley et al., 2014 and Ubezio et al., 2016) concerning the oscillatory behavior of endothelial cells in vascular tufts, however the amount of data provided are less than enough to make such associations.

Finally, the analysis of the neuronal compartment of the mouse retina is very limited, although it would be very beneficial for the impact of the report. Another application would be to skeletonizes larger portions of the vascular network (ultimately, the entire retina vasculature) in order to perform flow simulations. It would be interesting to understand the value of LSFM for those larger scale segmentations and the benefits for flow modelling.

Overall, this report shows that LSFM is an additional way to image the mouse retina. Nonetheless, authors failed to show that LSFM will bring strong benefits to our understanding of vascular biology when compared to existing methodologies. Thus, the impact and applicability of LSFM in this field is dubious.

[Editors’ note: further revisions were suggested prior to acceptance, as described below.]

Thank you for submitting your article "Mouse retinal cell behaviour in space and time using light sheet fluorescence microscopy" for consideration by *eLife*. Your article has now been reviewed by two peer reviewers, and the evaluation has been overseen by a Reviewing Editor and Anna Akhmanova as the Senior Editor. The following individuals involved in review of your submission have agreed to reveal their identity: Michael Dorrell (Reviewer #1); Kathryn Pepple (Reviewer #2).

The reviewers have discussed the reviews with one another and the Reviewing Editor has drafted this decision to help you prepare a revised submission.

In their article, Bentley and co-authors describe a novel use of LSFM for imaging and analyzing cellular and tissue morphology of mouse retinas during development and disease. As these models are frequently used to study retinal development and disease, as well as broader vascular disease, any model that improves upon current methodology will be highly beneficial. The improvements described for ex vivo live imaging, as well as co-staining and analysis of neuronal and vascular systems together are particularly impressive. The associated videos of endothelial dynamics were striking and of high quality. Adapting and expanding this technique to the visualization of additional intraretinal processes in ex-vivo samples holds exciting promise for the field of retinal biology and pathology. It is clear that the authors have made substantial improvements since the first version.

Essential revisions:

1) The shape of the 3D tufts is interesting, but the major use of the OIR model is to quantify changes in neovascularization, usually in response to gene alterations, protein agonists or antagonists, or various drugs. Certainly, differences in tuft size are observed by traditional confocal analysis and quantifiable changes have been made for decades with several novel adaptations over the years. The authors should compare the two methods in a study testing a drug or transgenic mouse known to affect tuft formation to see if there is indeed a dramatic improvement in quantification analysis in order to compare the two methodologies for this model system. While the structural analysis of the tufts (spirals, tuft fusion, etc.) are a very interesting finding, in terms of a broad improvement to the model, the benefits towards the OIR model's major purpose are less clear. If the authors choose not to perform the comparisons, the claims of their method being "better" would need to be appropriately tempered throughout the article.

2) The PACT analysis and description is good, along with the rational for choosing PACT. However, it is unclear if the authors ever tested that technique on transgenic mice where the fluorescence is innate to the tissue (not stained). Was the PACT procedure used for the live retinal vasculature studies in Figure 4? As one of the key parameters for choosing PACT, it seems that the maintenance of fluorescent protein emission should be tested and demonstrated using this procedure. Also, are clearing methods required for the vascular studies, or just for the neuronal studies and assessing retinal layers?

3) While the authors do a good job of demonstrating observations of subcellular structures like Golgi and actin dynamics, the authors should better discuss what might be learned from this level of resolution. For example, the authors mention using Golgi localization to study EC polarity in migration and regression, but the images shown in Figure 1—figure supplement 1A make it seem like any such interpretation would be difficult. Similarly, it is unclear what is shown with regards to the actin dynamics in Figure 4 and corresponding videos or the practical utility of such studies.

4) The data in Figures 5 and 6 are difficult to interpret in isolation and the 3D rotational videos only provided limited assistance. Ideally to support the morphologies proposed in the schematics in Figure 6C, F, and I, additional stains or some other form of validation of the author's interpretations could be provided (histology, perfusion/fixation?). Particularly if the "previously unappreciated knotted morphology of the tuft was evident across all tuft classes" is a truly significant and novel advance to the field. If no new data are added, it would be good to tone down the writing.

5) Current key limitations of flat mounting and confocal imaging are live imaging capabilities and quantifying neuronal cells like photoreceptors in models of degeneration (including the OIR model). The authors show great improvements in live imaging and 3D neuronal staining in the spherical eye cups. If the authors can show quantification of photoreceptors (or other neurons) in a degenerative model, that could greatly increase the applicability of the model system.

---

## [Author Response]

[Editors’ note: the authors resubmitted a revised version of the paper for consideration. What follows is the authors’ response to the first round of review.]

Reviewer #1:[…]Overall, this work is of high quality and demonstrates that the application of LSFM microscopy will be a useful technique to study angiogenesis especially when improved 3D resolution is warranted. The potential for LSFM in live imaging appears to be limited in spatial resolution and time.

We thank the reviewer for their positive response. To better assess the temporal/spatial resolution of live imaging we have now performed a much larger live imaging study, repeating imaging across spatial scales, thereby extending on the single examples given of WT and OIR in the previous version. Thereby, we have included more n and also show live imaging of subcellular actin dynamics using lifeact mice. Now our live imaging sample size is n=10, comprised of: mT/mG WT n=3, lifeact WT n = 6, mT/mG OIR n=1).

Furthermore, we found LSFM live imaging to be genuinely reproducible as we performed it across three different country locations with different collaborator teams to obtain the complete set (due to my own lab moving from the US to Sweden during the revisions and no longer having a mouse colony combined with technical issues with the Lightsheet microscope in Sweden). We have also developed a new method that limits curling up of the eye tissue permitting a wider field of tissue that can be live imaged (described in the Materials and methods: “To minimize curling of the retina, apply 1-2 drops of low melting agarose on retina and start uncurling the retina before the gel is solidified. It can then be transferred to the cylinder for imaging.”).

We find the approach is best suited to capture fast processes, given the high frame rates achievable with the LSFM over confocal. Altogether, the new cohort of data indicates the reverse of the reviewer’s comment – we do not find LSFM spatially restricted – subcellular processes across wide fields of view are entirely feasible, and we find high temporal resolution is also reliably achievable. The limitations rather lie currently with the overall length of a live imaging study (~2 hours with confidence). However, with a full optimization study of the tissue culture medium and conditions it may well be feasible to extend this window in the future, as the neurons in retinal explants are known to stably persist and grow in culture. The conditions used here may simply be too normoxic, to promote further angiogenesis.

At this point I have three concerns:1) In Figure 3 the authors perform comparative analyses of a sample image acquired by LSFM and confocal microscopy. The confocal image (Figure 3A) appears to be saturated, which makes quantitative analysis difficult. In general, the authors should provide histograms to ensure that comparable images are being analyzed. Also, the pixel depth of images should be specified.

We thank the reviewer for this important comment. We have now removed the original Figure 3 and the entire section on automated image analysis of vessel diameters as we discovered some calibration issues and artifacts with the meshing software (holes and complications in the mesh). The software is undergoing a more thorough development and a refinement process before we publish it separately in the future.

However, to clarify, we apologize for the confusion over what the original images show in Figure 3. The images were not saturated as they were all the binary images from the thresholded original images. We are sorry this was unclear. The original image, binary image and histogram of the number of saturated pixels in each image is shown in Author response image 1 for posterity.

**Author response image 1. respfig1:** Representative of original maximum intensity projection (MIP) of IsoB4 stained vessels (**A**), followed by thresholding to make a mask of vessels (**B**). Histogram of the original image showing the images are not saturated (**C**).

Generally, most of the data presented are rendered images. A better representation of less processed images would be helpful.

We appreciate the reviewer’s comments and have now included several unrendered images, for example Figure 1—figure supplement 1 and Figures 4, 6B and 7.

*2) When considering alternatives to confocal imaging, e.g. live imaging. The authors should also discuss optical coherence tomography (OCT), which has also been successfully applied for* in vivo *imaging of the mouse retina.*

The original section on OCT we agree was short and only focussed on prior live imaging work. Our reasoning for this is that we are primarily concerned with fluorescent imaging approaches that can provide molecular mechanism detail. But realize this needed to be made clearer. We have now included the following:

“Optical Coherence Tomography (OCT) is an established medical imaging technique that uses light to capture micrometer-resolution, three-dimensional images non-invasively, now widely used as a diagnostic tool (Srinivasan et al., 2006; Huber et al., 2009). […] Furthermore, being a non-fluorescent method, specific proteins cannot be labelled and tracked to investigate mechanism.”

3) To assess the spatial resolution of LSFM compared to other imaging techniques it would be helpful to examine finer morphological structures – for example cell junction, Golgi or mitochondria.

We have now included LSFM images of retinas showing Golgi, collagen and F-actin (lifeact), showing that these finer morphological structures can be successfully imaged with LSFM (Figure 1—figure supplement 1C-I).

Reviewer #2:[…] A critical aspect of this manuscript is the value-added of this new technology. As an outside evaluator, I see both positives and negatives of this technology and it would be critical for the authors to provide a more balanced view of these.

We thank the reviewer for the positive and supportive comments on this study. To provide a clearer comparison of the pros and cons of both LSFM and confocal imaging of mouse retinas we have significantly extended the previous short Discussion section on into a full balanced Discussion. We also have now emphasized the inclusion of this clearer, balanced comparison to confocal in the Introduction, Abstract and Discussion, e.g. In the Abstract: “We compare our results to standard retinal imaging methods, in particular confocal microscopy. Through quantitative correlative Confocal-LSFM imaging we find that flat mounting retinas for confocal microscopy significantly distorts tissue morphology.”

Or perhaps, if I am missing something, explain to the review team what is the misinterpretation on the aspects of the study indicated below.1) A detailed evaluation of the images provided reveals the lack of a clear hierarchy when retinas are evaluated by LSFM versus confocal flat-mounted images. For example, Figure 1F shows the vasculature of a P10. At this time, the vessels should clearly depict hierarchic branching patterns with alternating arteries and veins. Instead, the image shows a mesh or vessels that lack hierarchical organization and absence of alternating arterial-vein vessels. Is this outcome due to the fact that three vascular layers are juxtaposed, providing excessive noise to the inner layer (the hierarchical layer)? If this is the case, there are several negatives associated with this approach, although again, there are several positives (important to bring balance to the manuscript).

The vessels shown in Figure 1F are imaged from the outside of the eye, so you predominantly see the deeper vascular plexus, which has less hierarchical structure at P10 than the inner layer. This is not easily studied using confocal so represents a benefit if the deeper plexus is the object of the study. However, if the eye is rotated relative to the objective such that we view from above/inside – the superficial layer is closest and cross-sectional structures can be seen straight on. However, the reviewer is correct; if an older eye is being imaged the three layers will be somewhat visually overlapping (e.g. in Figure 1—figure supplement 1B), which could be hard to manually untangle due to the curvature, and as such represents a limitation. The preservation of the tissue depth information in the large z stack, however, means by computationally fitting to the local curvature of the eye tissue one could computationally colour code and subtract the retinal layers out for independent viewing and analysis. We have added a note on this to the balanced Discussion section.

2) For the visualization shown in Figure 3: Was this done using a flat mounted image? Or Having the eyeball as a curved structure? I believe it might be the former, and in this case, it is difficult to understand why there are differences in the evaluation. The authors offer that assessment of the vasculature using lighsheet shows wider vessels than confocal, but the problem with the out-of-focus light discussed by the authors brings a confounding factor on the metrics. Can the authors discuss this at greater length? Also how do these metrics compared to a perfusion-fixed transversal section of specific vessels?

We have now removed the original Figure 3 and the entire section on automated image analysis of vessel diameters as we discovered some calibration issues and artefacts with the meshing software (holes and complications in the mesh) so it is undergoing a more thorough development and a refinement process before we publish it separately in the future.

To answer the reviewers first question here and to better explain the tissue handling and correlative imaging process we added schematic Figure 3A. Indeed, the tissue was imaged curved first in LSFM and then flat mounted to view the same regions in confocal to compare any distortions directly.

To be completely confident of the comparative measurements we had two independent postdocs hand measure the width and depth of the same vessel segment in the corresponding images. We also re performed the analysis on LSFM vessels after deconvolution using Huygens but found no differences to the measurements made with the unprocessed LSFM images. Thus, to answer the reviewers second question we can confirm that out-of-focus light did not affect the measurements in these images. We suspect this is due the early postnatal stage of these retinas (P4) meaning they are very thin so have minimal light scatter.

To address the final point, we scoured the literature but could not find corresponding measurements of these very small vessels that we analyzed using perfusion-fixed transversal sections, but would be very open to comparing if a reference could be highlighted?

3) It is hard to interpret Figure 4 without a full legend (missing in the manuscript) but the comparison between confocal and LSFM clearly shows the advantage of LSFM, as the tufts are not squeezed through mounting. However, I am having a hard time in really understanding what is a "mature" versus a "knotting" tuft. If a nomenclature is to be introduced, then a more detailed description, assessment of multiple retinas, etc should be provided.

We apologize entirely for the lack of figure legend to Figure 4 (now Figure 5). This was a great disappointment as this figure really does indeed showcase one of the major benefits of LSFM as the reviewer points out – improved imaging and understanding of vascular malformations. Indeed, we have included quantification of the exact level of distortion confocal flatmounting incurs to tufts (Figure 5E).

The analysis was performed over multiple retinas, n=6 and each dot in the quantitative analysis (Figure 5B-D) is an individual tuft analysed.

To better explain the knotting nomenclature in response to the reviewer, we have now included a much deeper study of the 3D morphological structure (Figure 6), we explored deconvolution (reducing lightscatter using Huygens software) and using different 3D methods to view/understand, draw and analyse the 3D structures. This extra effort has turned out to be transformative, revealing nuances of complex vessel interlacing structures not before reported.

To simplify the nomenclature, we have changed to ‘small, medium and large’, instead of ‘initiating, knotting and mature’ as per reviewer 3’s request to avoid inferring too much temporal ordering of the process from static images until we know more. However, we make it clear in the discussion our assumption is that this is the likely ordering, based on the peak proportion of curved nuclei in medium sized tufts.

4) Clearly a big advantage of LSFM is the live imaging shown in Figure 5. Because there are multiple image modalities, it will be important to clearly state how is the ex-vivo preparation (flat? Curved? With media? Without? Heat? Oxygen?).

Information is in the Materials and methods section. In summary, Retina are curved in media containing 50% FCS with 5% CO2.

Reviewer #3:[…]Authors start by claiming that visualizing blood vessels in the retina's natural spherical conformation will be important for our understanding of vascular biology. Thus, a central point is to compare this new method with already existing methods, such as confocal microscopy. The only comparison done between both is related to vessel diameter, and no changes were found.

We agree and have committed to a major overhaul of the paper to strengthened and improve the comparison to confocal imaging throughout the paper. The revised version now includes many new quantifications, including comparative OIR tuft depth distortions across tuft sizes when imaged with confocal vs LSFM (Figure 5E); a substantial new vessel diameter analysis, performed by hand by two independent postdocs, following a correlative imaging approach to ensure precision and accuracy, that the exact same vessel diameter is directly compared between the imaging modalities. This study revealed that distortions are present in even very small developing vessels when imaged with a confocal (Figure 3). Furthermore, we have added a significant new section to the Discussion comparing the benefits and limitations of LSFM and confocal as suggested by reviewers 2 and 3.

Other quantitative information used in vascular biology includes: vascular density; number of branching points; number of endothelial tip cells; number of filopodia per tip cell; number of regressing branches; etc. Does LSFM improve the analysis of those parameters? It is likely not the case.

In the new Discussion section we point out that for such static quantifications of XY plane properties of developmental angiogenesis in just the superficial plexus, which all these quantifications relate to, confocal has better resolution and remains preferable to LSFM. We are not proposing LSFM to replace for this type of analysis, and go further to demonstrate with LSFM that the vessels in this superficial layer during development are not distorted, adding further confidence to the confocal approach in this setting. However, for pathological angiogenesis, where enlarged or abnormally oriented vessels grow, e.g. as in the tufts that protrude outside of the superficial plexus into the vitreous, we find a distinct advantage of LSFM over confocal. The tuft morphology is much less flattened and distorted, and the ability to image all around them and rotate in videos to view the tuft fully from the side or any angle helped to reveal the complex knotted morphology, not previously discovered across the many confocal studies of the OIR vasculature.

The key strength of LSFM is for rapid 3D-4D imaging, which can reveal many other important regions/processes in the retina that currently are less well studied or standardized, in vascular biology and opthalmology, beyond developmental angiogenesis in the superficial plexus. Analysis of deeper layers, diving vessels, interconnecting neurons and vessels with differential orientations in the tissue, abnormally enlarged or altered 3D structures such as vascular tufts, but potentially any region of the eye that has become abnormally shaped in 3D, might benefit from this type of imaging.

Moreover, description of the deconvolution and segmentation parameters in Materials and methods is very vague and relies heavily on "experts" (subsections “Manual cropping of well-segmented regions” and “Isosurface extraction”).

We have entirely removed the automated analysis pipeline which this comment relates to, while it undergoes more rigorous development and calibration due to some meshing artefacts that may have affected results. We have replaced the Materials and methods section on deconvolution, which was subsequently performed using Huygens software in this revised version, with a more detailed description of the process (Materials and methods).

Another advantage of LSFM could be that the new method is faster when coming to perform the same task (extract quantitative information of fluorescence images). However, authors did not discuss this aspect. It would be important therefore to include the amount of time needed to image and quantify these parameters per retina and compared it to confocal microscopy.

We agree with the reviewer, and have emphasized better the speed improvements with LSFM, indeed putting it as the first benefit listed, in the new Discussion section comparing LSFM and confocal. Generally speaking, while the image capture is dramatically faster for LSFM, we found that the quantification time (the time required to quantify the images) is comparable between LSFM and confocal, which has also been added to this section.

Authors also showed the capacity to perform live imaging of endothelial cells in the mouse retina. Despite being technically possible, the reviewer is skeptical on the value of the information that can be obtained in this way.

In the original version submitted we had only attempted live imaging a small number of times, including one video of WT and OIR as proof of concept and agree this did not demonstrate the full potential, reproducibility or quantitative value of the approach. We have thus significantly improved the live imaging Results section, repeating the WT live imaging with mT/mG mice so n=3 and including a new study using WT lifeact mice n=6 showing subcellular actin dynamics can be quantified. Given cell shape, migration, division and junctional dynamics all depend on actin this potentiates novel studies with many mouse mutants to gain quantifiable results. Furthermore, we have now included quantifications of the filopodia dynamics comparing the healthy retina to the OIR, which already indicated that the filipodia are dramatically altered in the OIR condition, worthy of greater study and indicating that live imaging with LSFM could generate data with dynamic insight and value for development and disease models.

Given the tremendous change in the environment, including oxygen concentrations, loss of vitreous humor, lack of blood flow, loss of intraocular pressure, which are known to affect endothelial cell biology, it is debatable how reliable can be any information acquired with this protocol.

We agree there are of course aspects that have changed, however compared to the standard use of in vitro assays for live imaging endothelial behaviour, such as bead sprouting assays and ES cell assays the tissue environment and cell behaviour within the largely intact local retinal tissue around the vessels is likely to yield more realistic cell behaviour than in vitro. Furthermore, if studying tip cell dynamics and filopodia, these are not perfused vessels, so the lack of blood flow is not likely a huge factor. We also see this as a first step in a longer goal were we or other labs can improve the tissue medium, dissection process and exploring micro pumps and fluidics to perfuse vessels via the optic nerve to get closer and closer to imaging longer time windows with in situ processes in an otherwise fully working ex vivo mouse eye.

As a matter of fact, Sawamiphak et al., 2010 reported a similar protocol, but the impact in the field so far is very minor.

We discuss this particular study in the Introduction section of the manuscript and point out their protocol involves significant delay between dissection and imaging due to the need to flatmount the tissue first. The flatmounting also creates more distorting and destruction of tissue gradients than our protocol.

Moreover, no quantification of cell migration or filopodia dynamics was attempted in order to illustrate the potential value of the method. The same holds true in regard to the pathological OIR model.

We agree and this has been significantly improved with the addition of more live imaging repeats. We have now included a full quantification study of filopodia comparing vessels in the healthy tissue and OIR, as well as subcellular actin dynamics, overhauling and extending the two live imaging Results sections significantly as well as updating Figure 5 and including the new Figure 6.

A potential interest of the report comes from the analysis of vascular malformations, so-called vascular tufts, in the OIR model. Authors describe in high detail different vascular structures including tufts of different sizes, ranging from 2 to >20 endothelial nuclei, including the presence of some curved nuclei. Despite the novelty aspect of the observations, the reviewer is not convinced that the same structures could not be observed in confocal microscopy following careful analysis.

We have substantially increased the study of OIR tufts (Figure 6) in the revised manuscript to better demonstrate the importance of LSFM and hope it now convinces the reviewer that the due to undistorted, 3D rotational viewing of tufts that it enables we can perceive their knotted, swirling morphology. We now quantitatively show that confocal microscopy distorted tuft depth significantly meaning the side view of tufts is almost impossible to interpret and unreliable in confocal images (as shown in Figure 5), which means the same structures could not be observed in confocal following the standard flatmounting procedures.

Furthermore, it would be much more difficult to see curved nuclei using confocal microscopy due to the potential for distortion to be curving them and the lack of the extra perspective side view to tell if they are truly curved. We have extended a sentence on this in the results to better emphasize: “It should be noted that care should be taken to rotate the image stack to confirm nuclear curvature, as two nuclei parallel to each other can look like only one nucleus (Figure 5A, fourth panel row, blue arrow), emphasizing the importance of 3D imaging with LSFM as rotating and viewing tufts from the side without distortion is not possible with confocal.”

Within the same context, authors claimed "abnormal cell dynamics" in the context of pathological angiogenesis. However, the manuscript includes very limited data (one video on filopodia, and 2 videos on undefined vessels) and no quantitative information to back up authors' claims.

We have included data from 7 and 3 videos for healthy and OIR tissues respectively together with their quantifications in comment #1. My lab moved location during the revision process and we no longer maintained a mouse colony or had an O2 chamber. We were able to work with collaborators and image mT/mG and lifeact mice available in these labs but could not for practical reasons continue the OIR experiments.

They suggest that those observations confirm previous hypotheses (Bentley et al., 2014 and Ubezio et al., 2016) concerning the oscillatory behavior of endothelial cells in vascular tufts, however the amount of data provided are less than enough to make such associations.

We have removed this reference.

Finally, the analysis of the neuronal compartment of the mouse retina is very limited, although it would be very beneficial for the impact of the report.

Through a new collaboration with neuronal Opthalmology researchers at Schepens Eye Institute (now co-authors on the paper – Dong Feng Chen et al.) we were able to investigate the neuronal compartment further, resulting in new images of RGCs and vascular interactions, now Figure 2C and D.

Another application would be to skeletonizes larger portions of the vascular network (ultimately, the entire retina vasculature) in order to perform flow simulations. It would be interesting to understand the value of LSFM for those larger scale segmentations and the benefits for flow modelling.

We agree this is a very interesting and useful application of the approach outside of the current manuscript scope however we have included it in the Discussion and we have plans for a new postdocs project to be to develop imageJ plugins that will help with skeletonizing the entire, curved vasculature.

Overall, this report shows that LSFM is an additional way to image the mouse retina. Nonetheless, authors failed to show that LSFM will bring strong benefits to our understanding of vascular biology when compared to existing methodologies. Thus, the impact and applicability of LSFM in this field is dubious.

We have better summarized the advantages of using LSFM compared to confocal imaging in the new Discussion section as noted. We do feel the approach has broad benefits for eye research beyond vascular biology, but also that is has several key benefits for vascular biology in terms of studying abnormally enlarged or dynamic structures as described in response to point 1 above.

[Editors’ note: what follows is the authors’ response to the second round of review.]

Essential revisions:1) The shape of the 3D tufts is interesting, but the major use of the OIR model is to quantify changes in neovascularization, usually in response to gene alterations, protein agonists or antagonists, or various drugs. Certainly, differences in tuft size are observed by traditional confocal analysis and quantifiable changes have been made for decades with several novel adaptations over the years. The authors should compare the two methods in a study testing a drug or transgenic mouse known to affect tuft formation to see if there is indeed a dramatic improvement in quantification analysis in order to compare the two methodologies for this model system. While the structural analysis of the tufts (spirals, tuft fusion, etc.) are a very interesting finding, in terms of a broad improvement to the model, the benefits towards the OIR model's major purpose are less clear. If the authors choose not to perform the comparisons, the claims of their method being "better" would need to be appropriately tempered throughout the article.

The reviewer makes a very useful suggestion to better balance our claims of when exactly confocal or LSFM might be more appropriate given what the study might aim to quantify, e.g. as they note, in OIR studies. As this was a very interesting point, we took the reviewers advice and performed a comparative 3D analysis, LSFM study of a published OIR drug treatment (Yagasaki et al., 2014). We looked at the effects of Everolimus, an inhibitor of mammalian target of rapamycin (mTOR), previously analysed with standard confocal area metrics to see whether LSFM could a) match the confocal measures and/or b) improve upon the insights gained with 2D area measurements. We have added the following results to the text, Figure 6—figure supplement 3 and also added further discussion and clarification on when confocal is more appropriate for certain analyses such as 2D area measurements. Overall, we have also tried to temper the language and be more balanced throughout.

“Next, we investigated whether LSFM could provide added benefits for OIR drug study quantifications, when compared to confocal microscopy. […] Thus, we concluded that LSFM is more suitable for 3D volume and tuft morphology characterisation to understand the mechanism of action of OIR drug treatments than confocal microscopy.”

2) The PACT analysis and description is good, along with the rational for choosing PACT. However, it is unclear if the authors ever tested that technique on transgenic mice where the fluorescence is innate to the tissue (not stained).

Due to the high expression levels of mT/mG-YFP, and its proximity to the surface in these samples, PACT clearing of the tissue was not required. However, it was well established in the original PACT publication (Yang et al., 2014), and in many subsequent publications utilizing the PACT technique, that genetically encoded fluorescent proteins maintain their fluorescence throughout the PACT clearing protocol.

Was the PACT procedure used for the live retinal vasculature studies in Figure 4?

No, the PACT procedure was not used here. We have now indicated the images are of “uncleared” in the figure legend to clarify.

As one of the key parameters for choosing PACT, it seems that the maintenance of fluorescent protein emission should be tested and demonstrated using this procedure. Also, are clearing methods required for the vascular studies, or just for the neuronal studies and assessing retinal layers?

In general, the need for clearing increases with the developmental stage of the eye cup and the depth of the structure of interest within the tissue. Therefore, we only utilized clearing for the vascular and neuronal samples in Figure 2 and Figure 2—figure supplement 1. Currently, clearing of living tissue has not been demonstrated. Therefore, we were careful to calibrate our optical image system as close as possible to the refractive index of the tissue without impacting the viability of the living samples. This involved maintaining the sample in a physiological aqueous buffer, using a water dipping objective, and adjusting the Z-position of the light-sheet within the tissue to ensure optimal alignment with the focal plane of our imaging objective.

3) While the authors do a good job of demonstrating observations of subcellular structures like Golgi and actin dynamics, the authors should better discuss what might be learned from this level of resolution. For example, the authors mention using Golgi localization to study EC polarity in migration and regression, but the images shown in Figure 1—figure supplement 1 make it seem like any such interpretation would be difficult. Similarly, it is unclear what is shown with regards to the actin dynamics in Figure 4 and corresponding videos or the practical utility of such studies.

To address this concern, we have now improved on the subcellular section by performing a demonstration that LSFM allows for Golgi-nucleus polarity assignment. For this, we utilized a new Golgi and nuclear double reporter mouse (GNrep mouse) (Barbacena et al., 2019).

The following text has been added to the manuscript together with new in panels Figure 1—figure supplement 1E-H:

“Moreover, quantification of the nucleus-Golgi polarity axis was amenable when imaging the GNrep mouse (Barbacena et al., 2019), which expresses Golgi-localised mCherry and nucleus-localised eGFP upon Cre-mediated recombination, enabling visualisation of endothelial specific nuclei and Golgi apparatus. […] The ability to 3D rotate the undistorted vascular image stacks obtained with LSFM revealed hidden cells whose polarity could be analysed, not visible when analysing the same image stack using standard confocal 2D imaging (i.e. viewed only from above) (Figure 1—figure supplement 1F-H).”

We have also added further discussion on this, as suggested, to better explain the benefits that subcellular level LSFM studies of undistorted 3D eye tissue could bring:

“Subcellular level resolution in undistorted 3D retinal structures. […] However, we see the greatest potential for subcellular analysis in future studies analysing disruptions in cell polarity, or other processes at the subcellular level such as actin localisation in large pathological vessels or other retinal structures”

4) The data in Figures 5 and 6 are difficult to interpret in isolation and the 3D rotational videos only provided limited assistance. Ideally to support the morphologies proposed in the schematics in Figure 6C, F, and I, additional stains or some other form of validation of the author's interpretations could be provided (histology, perfusion/fixation?). Particularly if the "previously unappreciated knotted morphology of the tuft was evident across all tuft classes" is a truly significant and novel advance to the field. If no new data are added, it would be good to tone down the writing.

As suggested, we have now performed additional supportive 3D imaging using a different approach – we chose microCT imaging as it provides an independent high-resolution volumetric imaging of the undistorted 3D tissue structure with which to assess tuft structure and compare to LSFM. We found this feasible and assessed two retinas from control and OIR mice each finding that tufts indeed often had evident holes/invaginations and bridged connections to the plexus. We have added the following text and Figure 6—figure supplement 2:

“To further validate these unexpected tuft morphologies with an independent high-resolution 3D imaging method, we performed microCT on intact health control and OIR retinas. […] On close inspection we indeed found tufts also appear to have holes/invaginations (Figure 6—figure supplement 2B-C) indicating further study of these complex 3D structures is warranted.”

5) Current key limitations of flat mounting and confocal imaging are live imaging capabilities and quantifying neuronal cells like photoreceptors in models of degeneration (including the OIR model). The authors show great improvements in live imaging and 3D neuronal staining in the spherical eye cups. If the authors can show quantification of photoreceptors (or other neurons) in a degenerative model, that could greatly increase the applicability of the model system.

We appreciate the suggestion and have now included quantification of neuronal dropout in a Rho KO mouse model of neuronal degeneration when imaged with LSFM. The following text has been added to the manuscript and Figure 2—figure supplement 1.

“In order to establish whether LSFM could be used to quantify neuronal changes in a retinal degeneration model we imaged retinal cups from the Rho KO degeneration model (Figure 2—figure supplement 1) (Humphries et al., 1997). […] The ONL had almost entirely lost its stable convex curvature by 8 weeks in the KO retina and the inner nuclear layer (INL) also appeared ruffled when viewed in 3D which may be due to the unevenness of dropout of photoreceptors (Figure 2—figure supplement 1A, B).”